# Estimation of future rainfall extreme values by temperature-dependent disaggregation of climate model data

Niklas Ebers[1], Kai Schröter[2], Hannes Müller-Thomy[2*]

[1]Coordination Unit Climate and Soil, Thünen Institute, Brunswick, 38116, Germany
[2]Leichtweiß Institute for Hydraulic Engineering and Water Resources, Department of Hydrology, Division of Hydrology and River Basin Management, Technische Universität Braunschweig, Brunswick, 38106, Germany
*previously published under the name Hannes Müller

*Correspondence to*: Niklas Ebers (niklas.ebers@thuenen.de)

**Abstract.** Rainfall time series with high temporal resolution play a crucial role in various hydrological fields, such as urban hydrology, flood risk management, and soil erosion. Understanding the future changes in rainfall extreme values is essential for these applications. Since climate models typically offer daily resolution only, statistical downscaling in time seems a relevant and computational effective solution. The micro-canonical cascade model conserves the daily rainfall amounts exactly and with all model parameters expressed as physical interpretable probabilities avoids assumptions about future rainfall changes. Taking into account that short-duration rainfall extreme values are linked with high temperatures, the micro-canonical cascade model is further developed in this study. As the introduction of the temperature-dependency increases the number of cascade model parameters, several modifications for parameter reduction are tested for 45 locations across Germany are selected. To ensure spatial coherence with the climate model data, a composite product of radar and rain gauges with the same resolution was used for the estimation of the cascade model parameters. For the climate change analysis the core ensemble of the German Weather Service, which comprises six combinations of global and regional climate models is applied for both, RCP 4.5 and RCP 8.5 scenarios. For parameter reduction two approaches were analysed: i) the reduction via position-dependent probabilities and ii) parameter reduction via scale-independency. A combination of both approaches led to a reduction in the number of model parameters (48 parameters instead of 144 in the reference model) with only a minor effect on the disaggregation results. The introduction of the temperature dependency improves the disaggregation results, particularly regarding rainfall extreme values and is therefore important to consider for future studies. For the disaggregated rainfall time series of climate scenarios, an intensification of the rainfall extreme values is observed. Analyses of rainfall extreme values for different return periods for a rainfall duration of 5 min and 1 h indicate an increase of 5-10% in the near-term future (2021-2050) and 15-25% in the long-term future (2071-2100) compared to the control period (1971-2000).

## 1 Introduction

Climate change is an existential threat for humankind. Rising temperatures and changes in rainfall characteristics are globally projected, with severe regional impacts, resulting in an increased occurrence of rainfall extreme events (Gründemann et al.,

2022). Rainfall extreme values are required in many hydrological applications, e.g. for dimensioning purposes in engineering hydrology, soil erosion estimation (Pidoto et al., 2022), flood risk management (Viglione et al., 2010; Tarasova et al., 2019) and in urban hydrology (Ochoa-Rodriguez et al., 2015). Knowledge about future changes of temporal high-resolution rainfall extreme values directly relates to one of the twenty-three unsolved problems in hydrology described by Blöschl et al. (2019),

i.e. question 9: 'How do flood-rich and drought-rich periods arise, are they changing, and if so why?'). In particular, sub-hourly rainfall extreme events are important to analyse the number of pluvial floods, which are relevant for the mesoscale and finer, rather than fluvial floods, which are relevant for meso- and macroscale. In this context we expect that the introduction of temperature dependency in rainfall disaggregation improves the representations of sub-hourly rainfall extreme values in climate change projections.

To prepare for future climate conditions the Intergovernmental Panel on Climate Change (IPCC) introduced the "Representative Concentration Pathway" (RCP) – climate scenarios (IPCC, 2014). These scenarios are based on different evolutions of the radiative forcing in the 21st century. In the sixth assessment report the IPCC established the "Shared Socioeconomic Pathways" (SSP) – scenarios representing a range of social-economic trajectories into the future. The Coupled Model Intercomparison Project (CMIP) coordinates the climate model simulations globally and is responsible for the new

climate model generation of the sixth phase (CMIP6) in which the RCP and SSP scenarios are combined. However, for the CMIP6 there are no bias-adjusted and regionalized climate model data available for Germany so far, but for CMIP5 these climate model data exist. Therefore, CMIP5 is considered as 'state-of-technic' climate simulations for Germany in this study. In CMIP5 the RCP scenarios are provided as part of the external forcing to the Global Climate Models (GCMs), which simulate climate projections on the global scale The spatial resolution of GCMs with ~150 km (Taylor et al., 2012) is limited by

computational capabilities, and is too coarse for hydrological applications on the micro- and meso-scale. To increase the spatial resolution, the outputs of GCMs serve as input for Regional Climate Models (RCM), which simulate the atmospheric conditions on a finer spatial resolution for a smaller extent. For Europe, the EURO-CORDEX initiative (driven by CMIP5) provides the results of several RCMs with a spatial resolution of ~50 km or ~12.5 km and a temporal resolution from 1 h to seasonal means.

The coarse temporal resolution of RCMs is a problem, limiting the study of rainfall extreme values to a coarse spatial scale. Many studies (e.g. Al-Ansari et al., 2014; DeGaetano et al., 2017; Araújo et al., 2022) only consider daily rainfall time series to evaluate changes in future rainfall extreme values. However, daily time series are insufficient for processing many hydrological applications. Berne et al. (2004) identified for urban catchments (1-10 km²) a minimum temporal resolution of about 3-5 min to model rainfall–runoff dynamics adequately Analysing various combinations of temporal (1–10 min) and

spatial (100–3000 m) resolutions for different urban catchments, Ochoa-Rodriguez et al. (2015) identified for an urban drainage area >100 ha a spatial resolution of 1 km and a temporal resolution of at least 5 min as minimum. Ficchi et al. (2016) investigated the influence of temporal resolution on streamflow simulations over a large and varied set of 240 mesoscale catchments (average catchment area 356 km²) and 2400 flood events. The input rainfall time series had a temporal resolution of 6 min to 1 day. They found that rainfall time series with a fine temporal resolution significantly improved the streamflow

simulations. On average, the best improvement across all 240 catchments was obtained with a 6 h resolution. The simulation of flood peaks and timing improved with increasing temporal resolution, highlighting the need for high-resolution rainfall time series and thus rainfall extreme values.

    Hence, for many hydrological applications sub-hourly rainfall extreme values are required, which is in contrast to the available temporal resolution from the RCMs. To overcome this issue, various methods exist to generate rainfall time series with a finer
temporal resolution from climate scenario data.

    Possible solutions are either the generation of rainfall time series with statistical input from the climate model data, e.g. delta change approach (Michel et al., 2021; Navarro-Racines et al., 2020), physical-based statistical methods (Marra et al., 2024) or the temporal disaggregation of future rainfall time series. The advantage of rainfall disaggregation is that the disaggregation model parameters can be estimated from the observed high-resolution rainfall time series, which ensures a correct
representation of time series characteristics. Well-known disaggregation methods are the method-of-fragments (Westra et al., 2012), Bartlett-Lewis rectangular pulse model (Koutsoyiannis and Onof, 2001; Onof and Wang, 2020) and cascade models (Molnar and Burlando, 2005; Paschalis et al., 2012; Müller and Haberlandt, 2018; Derx et al., 2023). An overview of different rainfall disaggregation methods is provided by Pui et al. (2012).

    Cascade models distribute the total rainfall amount of a coarse time scale (e.g. daily) on finer time steps (e.g. 5 min). A strong
advantage of the micro-canonical cascade model (Olsson, 1998) is that the rainfall amount of the coarser time step is conserved exactly at each disaggregation step, so aggregating the disaggregated time series results in the initial time series used for the disaggregation. The number of resulting finer wet time steps and their rainfall volumes depend on the so-called cascade generator. The required cascade model parameters are estimated from observed time series with the desired temporal resolution. Since for the method-of-fragments the amount of fragments is limited by the observation length (critical especially
in combination with temperature-dependency), and for the Bartlett-Lewis rectangular pulse model assumptions about distribution function of pulse characteristics have to be made, the cascade model was chosen for this study. The only assumption for the application of cascade models for the disaggregation of future climate model data is that the scaling behaviour of rainfall remains stationary, which is not questioned to the authors knowledge.

    In this study, a micro-canonical, multiplicative cascade model is used to achieve a final resolution of 5 min (Müller and
Haberlandt, 2018).

    The Clausius-Clapeyron relationship describes the temperature dependency of heavy rainfall events (e.g. Allen and Ingram, 2002). An increase of 1 K in air temperature causes an increase of the maximum possible air moisture content in the atmosphere and hence precipitable water amount about 7 %. However, the future increase in the occurrence of rainfall extreme events is highly non-linear and likely to be higher than 7 % per 1 K, but will vary regionally as it depends strongly on regional warming
(Seneviratne et al., 2021).

    Bürger et al. (2019) showed that the future change in sub-hourly extreme rainfall events depends on local temperature. Bürger et al. (2021) applied a temperature-dependent, simple canonical multiplicative cascade model to analyse the future change in rainfall extreme values with a temporal resolution of 10 min for stations in Germany, Austria and Switzerland. Bürger et al.

(2021) were able to determine a positive trend in the exceedance counts of rainfall events larger 5 mm/ 10min and a return

period of three years, which can be well explained by the climate change-related increase in temperature.

An introduction of temperature-dependency will lead to an increase of cascade model parameters. To keep the cascade model as parameter parsimonious as possible, several approaches for parameter reduction are analysed in this study. One possibility for parameter reduction is taking advantage of the scale dependency of the cascade model parameters. Assuming scale invariance, the same set of cascade model parameters is used for several disaggregation steps. Olsson (1998) tested before an

averaged parameter set estimated from coarser resolution (17h aggregated up to 5.7 days) for the disaggregation of time steps from ~17h to ~1h and found only a minor worsening of the disaggregation results. Olsson (1998) also showed that parameters estimated from time series with a temporal resolution <1 h differ from parameters estimated from a time series with a coarser temporal resolution. This is confirmed by studies of Güntner et al. (2001) and Rupp et al. (2009). Veneziano et al. (2006) analysed the mono-fractal scaling behaviour and identified two scaling ranges from daily to hourly resolution and from hourly

to 5 min resolution. In addition, Pöschmann et al. (2021) analysed the temporal scaling behaviour of extreme rainfall in Germany identifying three scaling ranges with approx. 1 h and 1 d as boundaries.

Another option for parameter reduction are so-called intra-event similarities (e.g. Willems et al., 2012), which describe parameter similarity of position classes applied for the cascade model. A detailed description of the intra-event similarities can be found in Sec. 3.2 to avoid technical details in this section.

The impact of both, the introduction of temperature-dependency and different approaches for parameter reduction, on the generation of sub-daily rainfall extreme values is analysed in this study. The following research questions are examined in detail:

i)    How can a cascade model been modified to improve the disaggregation results regarding rainfall extreme values with a minimum increase in model parameters?

ii)    How will rainfall extreme values with a temporal resolution of 5 min change in the future across Germany?

The paper is organized as follows: In Section 2 the study area, the rainfall data and the used climate scenario data are described. The applied methods are explained in Section 3, with the disaggregation model and its parameter reduction in Section 3.1. In Section 3.2 the temperature dependency of the rainfall extreme values and its implementation into the cascade model are described. The daily minimum, average and maximum temperature are examined as external predictors. In Section 4 the

disaggregation results and the derived change of future rainfall extreme values from the RCP 4.5 and 8.5 climate scenarios of the German Weather Service (DWD) core ensemble are presented and discussed. Summary and outlook of the study are provided in Section 5.

## 2 Data and study area

### 2.1 Observed data

In this study two types of observed data are used i) temperature time series from recording stations and ii) radar-based rainfall data. The 45 analysed locations are located in Germany, Central Europe (Fig. 1). They represent a range of different climatic, meteorological and geographical environments.

The northern part of Germany is characterised by coastal areas and glacial shaped landscapes, resulting in low altitudes. The southern part is dominated by the Alpine mountains with altitudes up to 2900 m a.s.l.. In between, there are several

mountainous regions with altitudes up to 1000 m. According to the Köppen-Geiger climate classification, there are two main climate zones in Germany (Beck et al., 2018). The eastern part of the country is dominated by a cold climate (Dfb). The western part has a temperate climate (Cfb). Both climates are characterised by warm summers without a dry season.

Most locations have a mean annual rainfall amount of up to 750 mm, with larger rainfall amounts in summer (May - October). In the mountainous regions of Germany, rainfall amounts larger >1000 mm per year are observed. The flatlands in the northeast

have the lowest annual rainfall amounts in Germany. Locations in this area measured annual rainfall amounts of up to 500 mm only. Besides the spatial coverage the availability of a temperature time series for the same location was a second criteria. A subset of five representative locations (A-E) was selected to show some detailed results. Stations A-E are distributed across Germany and the locations differ in their annual rainfall amount and percentage split between summer and winter rainfall amount and are therefore representative for different climate zones in Germany.

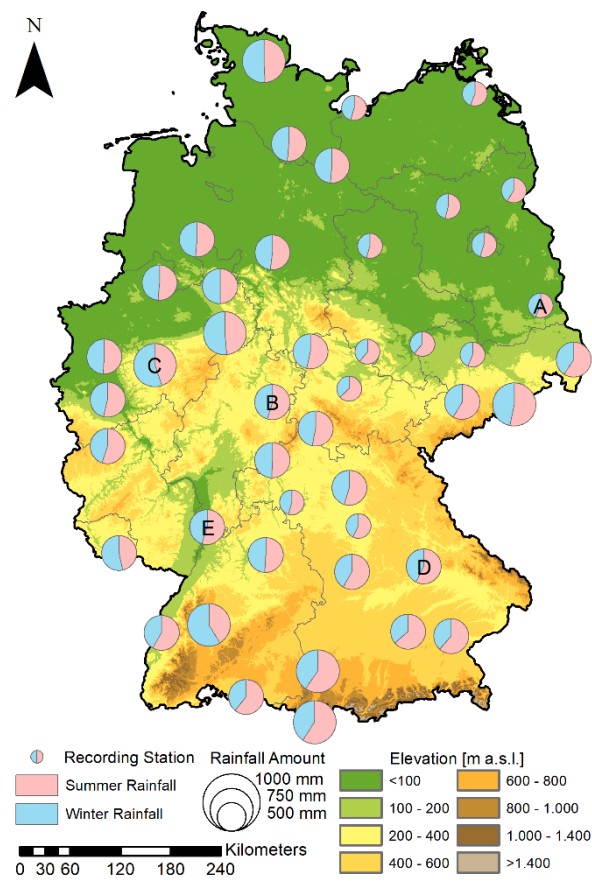

**Figure 1: Location of all recording stations (n=45) across Germany. Pie charts indicate the relative annual volume (radius) and percentage split between summer (red) and winter (blue) rainfall. Stations with letters represent the subset referred to in the method and result section (Source DEM: BKG).**

For the temperature time series at each location, data from recording stations are used. These recording stations are operated by the DWD and the observed time series are available as open-access (https://opendata.dwd.de/climate_environment/CDC/). The time series have a daily resolution. Measurements are operated following international standards two meters above the terrain surface. Available temperature data includes daily mean temperature, maximum temperature and minimum temperature. The temperature distribution in Germany depends on the distance to the ocean, elevation, latitude and season.

As rainfall data the YW-rainfall raster dataset (referred to as YW data from here) from the DWD with a temporal and spatial resolution of 5 min and ~1 km raster width is used. The YW data is based on a merged product of radar and rain gauge data for whole Germany (called RADOLAN) with hourly resolution, with subsequently disaggregation to 5 min time steps using the relative diurnal cycles from the radar. The quasi-gauge-adjusted YW data is available for the period 01.01.2001 – 31.12.2021 (Winterrath et al., 2018).

Table 1 provides an overview of station-based rainfall characteristics. Following the event definition of Dunkerley (2008) a rainfall event is defined as a rainfall period enclosed by at least one dry time step. A dry time step refers to a rainfall intensity

of 0 mm /5 min. The wet spell duration represents the duration of a rainfall event enclosed by two dry time steps. The wet spell amount is the sum of rainfall occurred during the wet spell. Dry spell duration is the duration of a dry period enclosed by wet time steps. During the pre-processing comparisons between the rain gauge time series and the YW data time series for the same location showed only negligible differences for the 5 min level.

165 **Table 1: Station-based rainfall characteristics for the observation period 2001-2021.**

| ID | Name | Altitude (m.a.s.l) | Mean annual precipitation (mm) | Average wet spell duration (min) | Average wet spell amount (mm) | Average dry spell duration (min) |
|---|---|---|---|---|---|---|
| 1 | Angermünde | 54 | 527 | 19.4 | 0.45 | 422.8 |
| 2 | Artern | 164 | 483 | 19.8 | 0.44 | 459.1 |
| 3 | Bamberg | 240 | 637 | 21.6 | 0.46 | 365.5 |
| 4 | Berlin-Tempelhof | 48 | 569 | 18.7 | 0.42 | 388.2 |
| 5 | Boizenburg | 45 | 642 | 19.6 | 0.43 | 326.8 |
| 6 | Boltenhagen | 15 | 597 | 19.9 | 0.44 | 374.2 |
| 7 | Chemnitz | 418 | 734 | 20.6 | 0.46 | 307.3 |
| **8** | **Cottbus (A)** | **69** | **563** | **19.3** | **0.42** | **380.1** |
| 9 | Diepholz | 38 | 694 | 20.3 | 0.46 | 341.0 |
| 10 | Düsseldorf | 37 | 755 | 20.8 | 0.51 | 363.8 |
| 11 | München-Flughafen | 446 | 751 | 23.9 | 0.54 | 362.0 |
| 12 | Erfurt-Weimar | 316 | 534 | 20.3 | 0.47 | 463.4 |
| 13 | Freudenstadt | 797 | 1555 | 26.3 | 0.69 | 227.7 |
| 14 | Gardelegen | 47 | 535 | 20.0 | 0.42 | 381.4 |
| 15 | Görlitz | 238 | 645 | 20.8 | 0.47 | 356.3 |
| 16 | Greifswald | 2 | 601 | 20.7 | 0.46 | 370.4 |
| 17 | Münster/Osnabrück | 48 | 733 | 18.9 | 0.45 | 338.5 |
| 18 | Hamburg-Fuhlsbüttel | 11 | 773 | 21.1 | 0.49 | 329.1 |
| 19 | Hannover | 55 | 628 | 18.8 | 0.44 | 368.7 |
| **20** | **Hersfeld, Bad (B)** | **272** | **657** | **19.0** | **0.42** | **327.7** |
| 21 | Kempten | 705 | 1233 | 26.4 | 0.63 | 251.3 |
| 22 | Kissingen, Bad | 282 | 669 | 19.8 | 0.43 | 338.0 |
| 23 | Köln-Bonn | 92 | 802 | 21.9 | 0.53 | 350.0 |
| 24 | Konstanz | 443 | 841 | 23.3 | 0.57 | 327.5 |
| 25 | Lahr | 155 | 712 | 22.1 | 0.56 | 367.2 |
| 26 | Leinefelde | 356 | 699 | 20.0 | 0.43 | 313.6 |
| 27 | Leipzig/Halle | 131 | 533 | 20.7 | 0.46 | 449.9 |
| 28 | Lippspringe, Bad | 157 | 900 | 21.4 | 0.49 | 280.4 |
| **29** | **Lüdenscheid (C)** | **387** | **1093** | **21.1** | **0.48** | **216.5** |
| 30 | Meiningen | 450 | 647 | 20.1 | 0.41 | 320.7 |
| 31 | Mühldorf | 406 | 817 | 23.4 | 0.52 | 328.5 |
| 32 | Neuruppin | 38 | 513 | 19.1 | 0.41 | 353.2 |
| 33 | Nürburg-Barweiler | 485 | 658 | 18.3 | 0.39 | 292.6 |
| 34 | Nürnberg | 314 | 604 | 21.9 | 0.48 | 396.8 |
| 35 | Oberstdorf | 806 | 1688 | 29.6 | 0.75 | 213.6 |
| 36 | Öhringen | 276 | 781 | 22.7 | 0.51 | 327.7 |
| 37 | Oschatz | 150 | 578 | 19.3 | 0.43 | 362.2 |
| **38** | **Regensburg (D)** | **365** | **656** | **21.9** | **0.46** | **344.7** |
| 39 | Saarbrücken-Ensheim | 320 | 867 | 23.3 | 0.53 | 302.6 |
| 40 | Salzuflen, Bad | 135 | 800 | 20.3 | 0.42 | 279.5 |
| 41 | Schleswig | 43 | 895 | 21.4 | 0.50 | 260.6 |
| 42 | Weißenburg-Emetzheim | 439 | 667 | 22.2 | 0.48 | 375.9 |
| 43 | Würzburg | 268 | 575 | 20.6 | 0.45 | 395.5 |
| 44 | Zinnwald-Georgenfeld | 877 | 1001 | 21.7 | 0.47 | 221.2 |
| **45** | **Mannheim (E)** | **96** | **638** | **22.0** | **0.53** | **418.2** |

## 2.2 Climate scenario data

In this study the RCP 4.5 and RCP 8.5 scenarios were analysed. The RCP 4.5 is an intermediate climate scenario, where climate
emission increase peaks in 2040 and declines afterward (Thomson et al., 2011). In contrast, the emissions for RCP 8.5 rise
throughout the 21st century. Each emission scenario provides part of the external forcing to the GCMs in CMIP5, which drives
the RCMs of EURO-CORDEX. The combination of a GCM and RCM creates one ensemble member of the RCP scenario.
EURO-CORDEX provides a variety of ensemble members with different combinations of GCM and RCM with a spatial
resolution of 0.11° and a minimal temporal resolution of 3 h.

In order to ensure a consistent data base for a variety of climate indicators and impact models the DWD selected six ensemble
members for each RCP, referred as DWD core-ensemble (Dalelane, 2021). In addition, the DWD core-ensemble is spatial
downscaled from 0.11° to a ~5 km raster. The downscaling process was carried out by the Federal Ministry for Digital and
Transport - Network of Experts Topic 1 (BMVI-Network of Experts Topic 1) using multiple linear regression (typical
distribution patterns of the respective climate variables served as predictors) and subsequent interpolation of the regression
residuals. The assumption was that a regional climate model correctly reproduces the coarse-scale patterns of the climate
variables to regionalize. Fine-scale structures of the respective climate variables are embedded in the regionalization process
by the typical patterns obtained from high-resolution reference data (Hänsel et al., 2020). The climate projections were bias-
adjusted before the spatial downscaling was carried out. This step was also undertaken by BMVI-Network of Experts Topic
1. For the bias-adjustment of the rainfall an univariate approach using the quantile-delta change mapping (QDCM) method
was chosen by Hänsel et al. (2020). The quantile mapping methods adapt the error-prone frequency distributions of the
projection data to those of real observations using an average mapping rule. However, the QDCM method is only applied to
rainfall values up to the 99.9th percentile, as rainfall amounts above the 99.9th percentile are not adequately represented in the
reference and projection data. For rainfall values above the 99.9th percentile, the adjustment value was extrapolated linearly.
For temperature data, a multivariate quantile mapping method was used. This method is an extension of quantile mapping, in
which, additionally to correcting the statistical moments it is also ensured that the consistency of the individual climatic
variables are maintained in relation to each other (e.g. relative humidity and air temperature) (Hänsel et al., 2020).

The temporal resolution of the DWD core-ensemble is daily. More information about the GCM-RCM-combination of the
ensemble members are provided in Table 2. The climate scenario data are available on request from the DWD as raster datasets
with a spatial resolution of ~5 km raster width, available for the period 01.01.1970 – 31.12.2100. Climate variables of the
RCMs used in this study are daily rainfall amounts as well as minimum, maximum and mean daily temperature. For the
analysis of the future change of rainfall extreme values, the C20 period (1971-2000) is compared with the near-term future
NTF (2021-2050) and the long-term future LTF (2071-2100).

**Table 2: Composition of GCM-RCM members of the DWD core ensemble for RCPs 4.5 and 8.5.**

| Ensemble member | RCP 4.5 | | RCP 8.5 | |
|---|---|---|---|---|
| | GCM | RCM | GCM | RCM |
| 1 | ICHEC-EC-EARTH (r1) | KNMI-RACMO22E | ICHEC-EC-EARTH (r1) | KNMI-RACMO22E |

| 2 | ICHEC-EC-EARTH (r12) | KNMI-RACMO22E | CCCma-CanESM2 (r1) | CLMcom-CCLM4-8-17 |
| 3 | ICHEC-EC-EARTH (r12) | SMHI-RCA4 | MOHC-HadGEM-ES (r1) | CLMcom-CCLM4-8-17 |
| 4 | MOHC-HadGEM-ES (r1) | CLMcom-CCLM4-8-17 | MIROC-MIROC5 (r1) | GERICS-REMO2015 |
| 5 | MPI-M-MPI-ESM-LR (r1) | MPI-CSC-REMO2009 | MPI-M-MPI-ESM-LR (r1) | UHOH-WRF361H |
| 6 | MPI-M-MPI-ESM-LR (r2) | MPI-CSC-REMO2009 | MPI-M-MPI-ESM-LR (r2) | MPI-CSC-REMO2009 |

## 3 Methods

### 3.1 Cascade model

A cascade model is used to increase the temporal resolution of a rainfall time series by distributing the rainfall amount of a coarse rainfall time step on finer time steps. This process is known as disaggregation. The number of resulting wet time steps and their rainfall amount depends on the cascade generator. The cascade model parameters are estimated from observed time series. The micro-canonical cascade model used for the disaggregation of daily time steps into 5 min intervals in this study was introduced by Müller and Haberlandt (2018, variant B2).

The branching number b indicates the number of time steps generated from the coarser time step and is therefore an important structural element of the model. In the first disaggregation step b is set to be 3 to generate three branches (b=3) of 8 h time steps (Fig. 2). For the following disaggregation steps, b = 2 is applied to generate two finer time steps from one coarser time step. In total, there are seven disaggregation steps to get from a daily time step to 7.5 min time steps. To generate 5 min time steps the rainfall amount of each 7.5 min time step is uniformly split into three 2.5 min time steps, with a subsequent aggregation of two non-overlapping time steps.

For the splitting with a branching number of b = 2, the weights $W_1$ and $W_2$ are used to distribute the rainfall amount from a coarser to two finer time steps. The sum of $W_1$ and $W_2$ is always 1 in each split, which conserves the rainfall volume exactly. This results in the following probabilities for W1 and W2 (Eq. 1):

$$W_1, W_2 = \begin{cases} 0 \text{ and } 1 \text{ with } P(0/1) \\ 1 \text{ and } 0 \text{ with } P(1/0) \\ x \text{ and } 1-x \text{ with } P(x/(1-x)); 0 < x < 1 \end{cases} \tag{1}$$

where P is the probability for of each combination of weights. A 0/1-splitting means that the entire rainfall amount is assigned to the second time step (W2 = 1), with no volume assigned to the first (W1 = 0). Vice versa, for the 1/0-splitting W1 = 1 and W2 = 0 are applied. A x/(1-x)-splitting distributes the rainfall volume over both finer time steps. The relative fraction x of the rainfall amount of the coarser time step assigned to the first time step is defined as 0 < x < 1. Considering x as a random variable for all disaggregation steps, an empirical distribution function f(x) is estimated from the observed time series (Müller and Haberlandt, 2018).

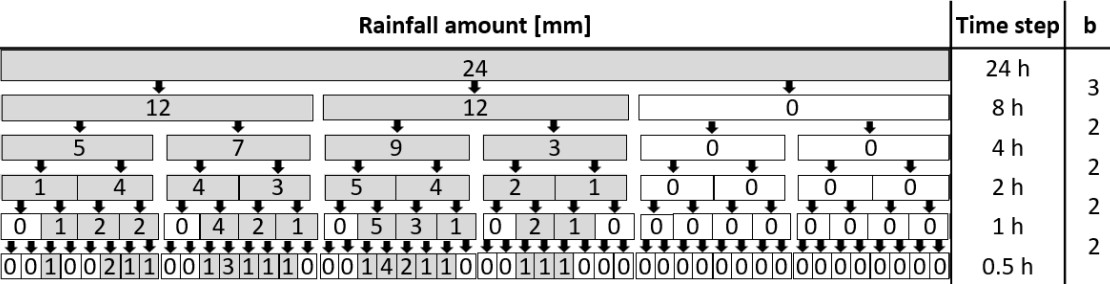

| Rainfall amount [mm] | Time step | b |
|---|---|---|

- 24h: 24
- 8h: 12 | 12 | 0
- 4h: 5 | 7 | 9 | 3 | 0 | 0
- 2h: 1 | 4 | 4 | 3 | 5 | 4 | 2 | 1 | 0 | 0 | 0 | 0
- 1h: 0 1 2 2 0 4 2 1 0 5 3 1 0 2 1 0 0 0 0 0 0 0 0 0
- 0.5h: 0 0 1 0 0 2 1 1 0 0 1 3 1 1 1 0 0 0 1 4 2 1 1 0 0 0 1 1 1 0 0 0 0 0 0 0 0 0 0 0 0 0 0 0 0 0 0 0

with b values 3, 2, 2, 2, 2 between levels.

**Figure 2: Multiplicative cascade model scheme for the first five disaggregation steps with branching numbers b, starting with a daily rainfall amount of 24 mm.**

The parameters of the cascade model are position- and volume-dependent. The position classes result from the wetness state of the current time step and its previous and subsequent time steps in the rainfall time series. The cascade model applied in this study has four position classes: starting, enclosed, ending and isolated. However, there are also cascade models that use less or more position classes (e.g. Rupp et al., 2009; Müller-Thomy, 2020) or different concepts as asymmetry (Maloku et al., 2023). A starting position class describes the first wet time step at the start of a rainfall event. Therefore, it is a wet time step

preceded by a dry time step and followed by a wet time step. An enclosed position class defines a wet time step surrounded by wet time steps. In contrast, an isolated position class is a wet time step between two dry time steps. The ending positing class describes a wet time step at the end of a rainfall event, which is preceded by a wet time step and followed by a dry time step. The volume-dependency of the parameters is considered by two volume classes with the mean rainfall intensity of a position class being an appropriate volume class threshold (Güntner et al., 2001). Each disaggregation step with a branching number

of b = 2 is represented by a parameter set consisting of three parameters (P(0/1), P(1/0), P(x/1-x)) for four position classes with two volume classes each. This results in a parameter sets with 24 parameters.

The cascade model parameters are scale-dependent. For each disaggregation step the model uses a single-parameter set (bounded cascade model). In contrast, an unbounded cascade model assumes scale-independency of the model parameter, whereby the same parameter set is applied over all disaggregation steps (Marshak et al., 1994).

The first disaggregation level with b = 3 requires more parameters than disaggregation steps with b = 2. From one coarse time step one, two, or three finer wet time steps can be created. This results in a large number of parameters if position-dependency is taken into account (Müller-Thomy, 2020). Hence, the disaggregation for b=3 is carried out without position-dependency, only volume-dependency is considered. The chosen threshold to distinguish lower and upper volume class is quantile q=0.998 of all positive rainfall amounts.

The parameters of the cascade model and f(x) are estimated for each location with 5 min time series for the period 01.01.2001 – 31.12.2021. The 5 min time series are extracted from a 5 km rainfall raster aggregated from the 1 km YW data. The aggregation of the YW data to a 5 km raster was used to ensure spatial consistency with the climate model data. Since the disaggregation is a random process, results vary depending on the initialization of the random number generator. Müller & Haberlandt (2018) found that after 30 disaggregation runs the mean value of the main rainfall characteristics did not change

significantly with an increasing number of disaggregation runs, hence 30 realizations are carried out for each analysis in this study.

## 3.2 Parameter Reduction

Temperature dependency will lead to an increase in the total number of cascade model parameters. To keep the cascade model as parameter parsimonious as possible, a reduction of the current number of cascade model parameters is studied.

Two approaches for parameter reductions are tested: based on i) scale invariance and ii) intra-event similarities. The reference model is a bounded cascade model with scale-independent model parameters (referred to S0). Assuming scale invariance of the model parameters, two different scaling ranges are analysed: 5 min to 1 h and 1 h to 24 h. For the parameter reduction, the disaggregation steps with b = 2 are suitable, since only these are applied over several scales. Based on the two scaling ranges, S1 represents an approach where one parameter set is used for each of the disaggregation steps from 8 h to 1 h and a second

parameter set from 1 h to 7.5 min. In approach S2 one single parameter set over all disaggregation steps from 8 h to 7.5 min is applied which corresponds to an unbounded cascade model.

Intra-event similarities are the ii) approach to reduce the number of parameters based on the position classes. The parameter with a probability $P_{starting}(0/1)$ describes the same event-cohesive systematic as $P_{ending}(1/0)$. The rainfall amount of a coarser time step is distributed so that a wet time step is not separated from an previous or following wet time step (Fig. 3A). A

connected disaggregation event can only occur if an enclosed time step ($Z_{1,3}$) is disaggregated with $P(x/(1-x))$ into two wet time steps. For an enclosed time step the probability of $P(x/(1-x))$ is very high. If a time step with a starting position ($Z_{1,2}$) followed by an enclosed position ($Z_{1,3}$) is disaggregated the probability for a 0/1-spliting is high, so that the rainfall event remains connected. The similarity of the probabilities $P_{starting}(0/1)$ and $P_{ending}(1/0)$ is also shown by the parameter values of the reference model (Tab. 3). A unification of the parameters in each volume class is therefore reasonable. Vice versa, the

parameters $P_{starting}(1/0)$ and $P_{ending}(0/1)$ are also similar (Fig. 3B). In Fig. 4, the probabilities resulting from the intra-event similarity for all 45 locations show only minimal changes (0-2 %) between the parameters for both similarities underlying our assumption. For similarity 1 the differences between $P_{starting}(0/1)$ and $P_{ending}(1/0)$ in $V_1$ are slightly higher compared to $V_2$. Contrary, in similarity 2 changes between the parameters are slightly higher in $V_2$. However, the difference is negligible. Model

variants where intra-event similarities are taken into account are referred to as P1 in Table 4. Model variants that do not consider parameter reduction via intra-event similarities are referred to as P0.

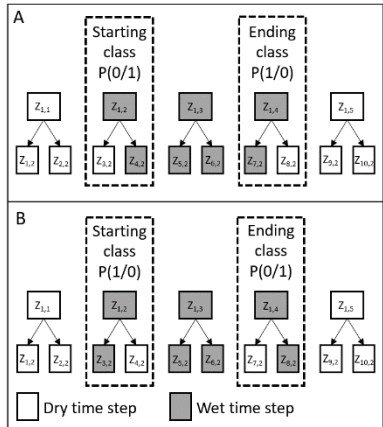

Figure 3: Disaggregation of a wet time step ($Z_{1,2}$ & $Z_{1,4}$) to describe the similarities of probability parameters in the start and end position class (for the same volume class) for continuous rainfall events (A) and non-continuous rainfall events (B).

Table 3: Comparison of the probabilities parameters [%] $P_{starting}(1/0)$ and $P_{ending}(0/1)$ for the two intra-event similarities approaches (Fig. 3 A & B) for both volume classes (V1, V2) at location A.

| Disaggregation step [h - h] | Similarity 1 | | | | Similarity 2 | | | |
|---|---|---|---|---|---|---|---|---|
| | $P_{starting}(0/1)$ | | $P_{ending}(1/0)$ | | $P_{starting}(1/0)$ | | $P_{ending}(0/1)$ | |
| | $V_1$ | $V_2$ | $V_1$ | $V_2$ | $V_1$ | $V_2$ | $V_1$ | $V_2$ |
| 8 – 4 | 34 | 62 | 31 | 61 | 2 | 13 | 3 | 13 |
| 4 – 2 | 32 | 62 | 34 | 60 | 5 | 12 | 2 | 14 |
| 2 – 1 | 30 | 60 | 27 | 60 | 5 | 18 | 6 | 15 |
| 1 – 0.5 | 30 | 60 | 30 | 60 | 1 | 9 | 2 | 10 |
| 0.5 – 0.25 | 28 | 60 | 25 | 61 | 2 | 11 | 1 | 12 |
| 0.25 – 0.125 | 25 | 58 | 25 | 58 | 2 | 9 | 2 | 10 |

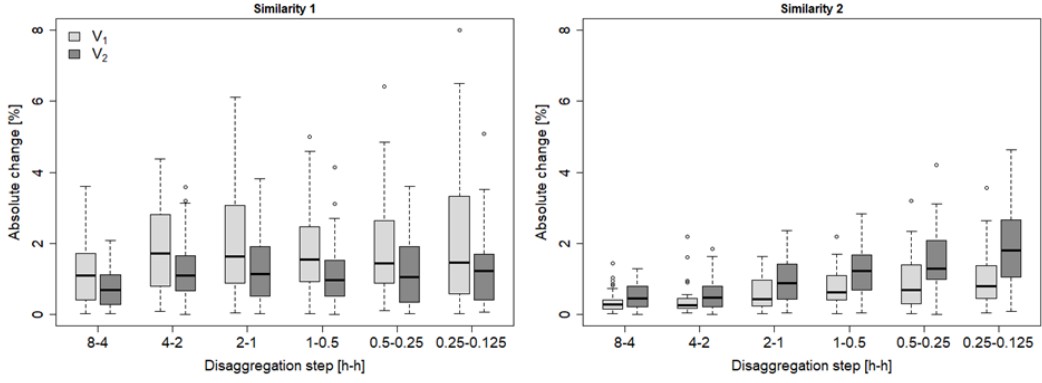

Figure 4: Absolute change [%] of the cascade model parameters from $P_{starting}(0/1)$ to $P_{ending}(1/0)$ (Similarity 1) and from $P_{starting}(1/0)$ to $P_{ending}(0/1)$ (Similarity 2) in both volume classes across all 45 stations and disaggregation steps.

A total of five variants of parameter reduction are analysed (Tab. 4):

S1-P0: Only the scale invariance of rainfall properties is considered. Therefore, unbounded parameter sets for the disaggregation 8 h to 1 h, and for 1 h to 7.5 min are applied.

S0-P1: Only the intra-event similarities are considered.

S1-P1: Combines the scale invariance of rainfall properties (S1-P0) and the intra-event similarities (S0-P1) resulting in two unbounded parameter sets.

S2-P0: One unbounded parameter set is applied for the disaggregation 8 h to 7.5 min.

S2-P1: Combines one bounded parameter set over all scales (S2-P0) and the intra-event similarities (S0-P1, lowest number of parameters of all variants).

**Table 4: Parameter composition for b = 2 splitting used in parameter reduction analysis.**

| Method Name | Parameter sets | Parameter number in parameter set | Total Parameter sum |
|---|---|---|---|
| S0-P0 (Reference) | 6 | 24 | 144 |
| S1-P0 | 2 | 24 | 48 |
| S0-P1 | 6 | 20 | 120 |
| S1-P1 | 2 | 20 | 40 |
| S2-P0 | 1 | 24 | 24 |
| S2-P1 | 1 | 20 | 20 |

## 3.3 Temperature dependency

A temperature dependency of the cascade model parameters is introduced to increase their physical background. First, the theoretical relationship between the temperature and rainfall extremes is reviewed for the station subset A-E. Since only daily temperature values are available from the climate scenarios, the dependency of 5 min rainfall intensities is analysed for daily data only. As predictors daily mean temperature, maximum temperature and minimum temperature are tested. All temperature characteristics were classified to estimate class-specific parameter sets. The class width was chosen so that each class contains a minimum of 10,000 time steps with rainfall intensities >0 mm/ 5 min, which leads to equidistant class widths of 5 °C. For class widths of 1 °C and 2.5 °C the number of included time steps per class were too small for some classes, precluding reliable statistical analysis. For each temperature class, the cascade model parameters are estimated separately.

## 3.4 Validation of the disaggregated time series

The disaggregated time series are validated regarding continuous and event-based rainfall characteristics as well as rainfall extreme values. Therefore, the relative error (rE) is used, which is calculated for all rainfall characteristics RC at each location i over all realisations n of the disaggregated (Dis) and observed (Obs) time series and then averaged over all stations (Eq. 2). In addition, the mean error (mE) is analysed, which is calculated from the difference between Dis and Obs at each location i (Eq. 3):

$$rE = \frac{1}{n} \cdot \sum_{i=1}^{n} \frac{(RC_{Dis,i} - RC_{Obs,i})}{RC_{Obs,i}} \tag{2}$$

$$mE = \frac{1}{n} \cdot \sum_{i=1}^{n}(RC_{Dis,i} - RC_{Obs,i}) \tag{3}$$

The rainfall extreme values are validated in two ways. First, the return periods of rainfall extremes values of the disaggregated time series are analysed. Therefore, empirical return periods (T) are estimated according to the German guideline DWA-531 (DWA-531, 2012):

$$T = \frac{L+0.2}{k-0.4} \cdot \frac{M}{L} \tag{4}$$

, where L is the number of rainfall events that is considered to be 2.4 time the length of the analysed time series number in years (M) and k the running index of the sample sorted by size. The rainfall intensities assigned to return periods $d_R(T)$ were identified with the exponential distribution function:

$$d_R(T) = u + w \cdot \ln(T) \tag{5}$$

, where u and w are parameters determined by linear regression plotting $d_R(T)$ against ln(T) using (4). The return period allows a validation of the most extreme rainfall events.

For the second validation the 99.9% quantile ($q_{99.9}$) of the disaggregated time series and the observed time series is applied. This criterion was used before by e.g. Bürger et al. (2021), Fumière et al. (2020), Myhre et al. (2019) and provides insight into the behaviour of the very high, but less extreme rainfall intensities.

## 325 **4 Results and Discussion**

### **4.1 Parameter reduction**

The impact of parameter reduction on continuous rainfall characteristics (rainfall intensity, dry spell duration, wet spell duration and amount) was analysed for the reference model S0-P0 and the five model variants listed in Tab. 4. The mean wet spell duration is underestimated by all model variants with a rE of about -23 % (Tab. 5). There is only a negligible difference

(<2 %) between the model variants and there is no impact of the parameter reduction approaches (intra-event similarities (P1) and scale invariance (S1, S2)) noticeable for the wet spell duration.

The mean rainfall intensity is overestimated in S0-P0 (rE = 24 %). The smallest deviations (rE = 19 %) are identified in S2-P0 and S2-P1. The wet spell amount is slightly underestimated by all model variants, with S0-P0, S1-P0 and S0-P1 showing the smallest deviation (rE = -4 %), while the largest deviation (rE = -9 %) is observed in S2-P0 and S2-P1. A noteworthy

aspect of the wet spell amount is the standard deviation of rE, which is with -11 % significantly larger for the approaches S2-P0 and S2-P1 than for the other approaches, with only -1 %. The dry spell duration shows a similar pattern compared to the wet spell duration with hardly any difference between the model variants. The mean rE is -13 %.

**Table 5: Relative Error (rE) [%] of continuous rainfall characteristics between disaggregated and observed time series for rainfall time steps > 0.1 mm (mean across 45 stations).**

| Rainfall characteristic | rE [%] | | | | | |
|---|---|---|---|---|---|---|
| | S0-P0 | S1-P0 | S0-P1 | S1-P1 | S2-P0 | S2-P1 |

| | | | | | | |
|---|---|---|---|---|---|---|
| *Wet spell duration [min]* | | | | | | |
| Mean | -23 | -22 | -23 | -22 | -23 | -25 |
| Standard deviation | -47 | -44 | -47 | -44 | -42 | -42 |
| *Rainfall intensity [mm/5 min]* | | | | | | |
| Mean | 24 | 23 | 24 | 23 | 19 | 19 |
| Standard deviation | 52 | 51 | 52 | 51 | 45 | 50 |
| *Wet spell amount [mm]* | | | | | | |
| Mean | -4 | -4 | -4 | -5 | -9 | -9 |
| Standard deviation | -2 | -1 | -2 | -1 | -11 | -12 |
| *Dry spell duration [min]* | | | | | | |
| Mean | -13 | -13 | -14 | -13 | -13 | -13 |
| Standard deviation | -27 | -26 | -27 | -26 | -22 | -22 |


The impacts of parameter reduction approaches on rainfall extreme values were analysed using the $q_{99.9}$ for each temperature class (Fig. 5) and for the two-year return period (Fig 6). The results show that all model variants tend to overestimate the $q_{99.9}$ rainfall intensity in the lower temperature classes (< 8–13 °C) while underestimating it in the highest temperature class (>18 °C). The relation between the $q_{99.9}$ rainfall intensity and the temperature classes is moderate for all model variants
compared to the observed data. Interestingly, station E shows no coherent relation between the $q_{99.9}$ and the temperature characteristics.

The parameter reductions mainly led to a reduction of the $q_{99.9}$ (Fig. 5). The model variants considering the scale invariance with two bounded parameter sets (S1-P0 and S1-P1) leads to slightly smaller $q_{99.9}$ than the reference model. However, the application of one bounded parameter sets (S2-P0 and S2-P1) leads to higher deviations from the reference model. The choice
of using one (S2) or two (S1) unbounded parameter sets has a higher impact on $q_{99.9}$ then the intra-event similarities.

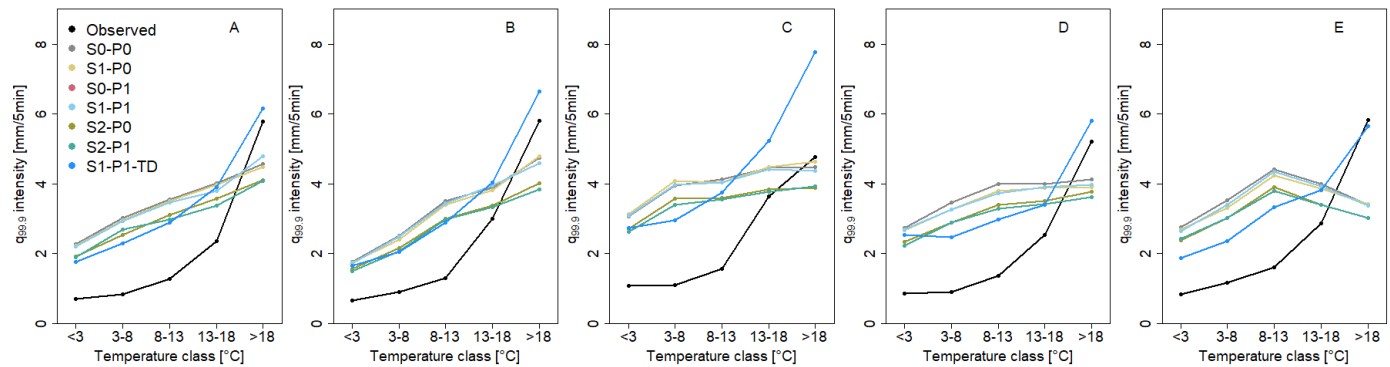

**Figure 5: Impact of temperature-dependency and parameter reduction on the $q_{99.9}$ rainfall intensity [mm/ 5min] for the disaggregated and the observed (quasi-gauge-adjusted radar data) time series in the temperature classes for selected locations (A-**
**E) and the daily mean temperature.**

For the two-year return period all model variants lead to overestimations of the extreme value resulting from the observations, with the reference model (S0-P0) showing a median rE of 40 % over all stations. There are only slight differences among the analysed parameter reduction approaches.

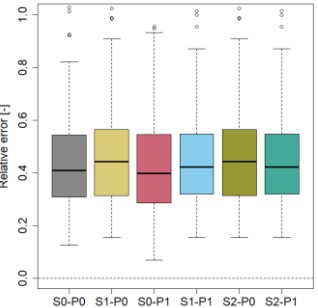

 **Figure 6: Relative error of the rainfall intensity with a return period of two years for all parameter reduction model variants over all locations.**

The parameter reduction based on the intra-event similarity had minimal effects on the continuous rainfall characteristics, $q_{99.9}$ and the rainfall extreme values, indicating that the underlying assumptions hold ($P_{starting}(0/1) \approx P_{ending}(1/0)$ and $P_{starting}(1/0) \approx$ and $P_{ending}(0/1)$).

However, the impact of the parameter reduction based on scale invariance was higher than on intra-event similarities. While approaches with two unbounded parameter sets (S1) showed almost no deviation from the reference model (S0), S2 with a bounded parameter set for 8 h to 7.5 min led to slightly different results for continuous rainfall characteristics and $q_{99.9}$. Differences include improvements (rainfall intensity and $q_{99.9}$ for lower temperature classes) and declines (wet spell amount and $q_{99.9}$ for higher temperature classes).

Since the overall aim of the first part of this study was to identify a parameter reduction without affecting the disaggregation results, approach S1-P1 combining the scale invariance of rainfall properties (S1-P0) and the intra-event similarities (S0-P1) with 40 parameters instead of 144 parameters is applied for the implementation of temperature-dependency.

## 4.2 Temperature Dependency

In Fig. 5 the positive dependency of $q_{99.9}$ on the mean temperature is clearly visible for the observed rainfall time series at the
locations A-E, indicating lower rainfall extreme values at low temperature classes. In addition, the difference of $q_{99.9}$ between the lower temperature classes were smaller, indicating a smaller temperature dependency for temperature classes <8 °C. This applies for all temperature characteristics. These findings are similar to Bürger et al. (2021) for 10 min rainfall data.
Although higher temperature classes are associated with higher $q_{99.9}$ values, the highest proportion of the 20 highest rainfall events is observed in the temperature class 13-18 °C (Tab. 6). Therefore, the rarest rainfall extreme events do not solely occur
in the highest temperature class. Across all stations, a larger proportion (20 %) of the 20 highest rainfall events falls within the

temperature class 8 to 13 °C, while only 15 % are in the highest class (>18 °C). Although rainfall extreme events at low temperatures are observed at some locations, e.g. location B, they remain exceptions.

**Table 6: Distribution of the 20 largest rainfall events among the temperature classes for the locations A-E and mean across all 45 locations.**

| Temp. Class | Proportion of rainfall extreme events [%] | | | | | |
|---|---|---|---|---|---|---|
| | A | B | C | D | E | ø 45 loc. |
| < 3°C | 0 | 5 | 0 | 0 | 0 | 0 |
| 3 – 8 °C | 5 | 0 | 5 | 10 | 0 | 5 |
| 8 – 13 °C | 10 | 35 | 25 | 20 | 15 | 20 |
| 13 – 18 °C | 55 | 60 | 60 | 70 | 60 | 60 |
| > 18 °C | 30 | 0 | 10 | 0 | 25 | 15 |


The implementation of temperature dependent parameters P(0/1), P(1/0) and P(x/(1-x)) resulted in a slight change of continuous rainfall characteristics for the daily mean, daily max. and daily min. temperature (Tab. 7). The overestimation of rainfall intensity was reduced from $rE = 23$ % (S1-P1) to rE 18-20 % (all S1-P1 with temperature dependency). The results for wet spell amount, wet spell duration and dry spell duration worsened slightly with 1-4 % for all S1-P1 with temperature

dependency.

Overall, the impacts of temperature-dependent disaggregation on the continuous rainfall characteristics are considered as negligible, with almost no differences among the three temperature characteristics.

**Table 7: Relative error rE of continuous rainfall characteristics of temperature-dependent and -independent disaggregated time series (mean for 45 stations) for S1-P1 (TDmin = daily minimum temperature, TDmean= daily mean temperature, TDmax = daily**
**maximum temperature).**

| Rainfall characteristic | Relative error rE [%] | | | |
|---|---|---|---|---|
| | S1-P1 | S1-P1-$TD_{min}$ | S1-P1-$TD_{mean}$ | S1-P1-$TD_{max}$ |
| *Wet spell duration [min]* | | | | |
| Average | -22 | -23 | -23 | -23 |
| Standard deviation | -44 | -40 | -39 | -40 |
| *Rainfall intensity [mm/5 min]* | | | | |
| Average | 23 | 19 | 20 | 18 |
| Standard deviation | 51 | 21 | 22 | 20 |
| *Wet spell amount [mm]* | | | | |
| Average | -5 | -9 | -8 | -9 |
| Standard deviation | -1 | -17 | -16 | -23 |
| *Dry spell duration [min]* | | | | |
| Average | -13 | -15 | -15 | -15 |
| Standard deviation | -26 | -24 | -23 | -22 |

To analyse the effects of the temperature-dependent disaggregation on the rainfall extreme values, the $q_{99.9}$ for 5 min rainfall intensities of each temperature class is evaluated over all stations (Fig. 7).

Without temperature-dependency (S0-P0 and S1-P1) the mE of $q_{99.9}$ exhibits a nonlinear pattern across the temperature classes.
The mE is roughly 1.5 mm/5 min for low temperature classes and increases to 2.3 mm/5 min for medium temperature classes. The mE then drops significantly to -1 mm/5min in the highest temperature class. Thus, $q_{99.9}$ is underestimated at high temperatures in the absence of temperature-dependency.

If the temperature-dependency is taken into account (S0-P0-TD and S1-P1-TD), the pattern of the mE changes. In general, a slight negative relation for mE is observed. These changes are observed for all temperature characteristics (min., mean and
max.), with minimal differences between them. For the mean temperature (Fig. 7b), the mE decreases by approximately 0.75 mm/5min compared to the model variants without temperature-dependency in the mean temperature classes (3-8 C° and 8-13 °C). The highest temperature class (>18 °C) shows the greatest effect of temperature-dependency, with an mE increase of 1.5 mm/5 min, leading to a slight overestimation of $q_{99.9}$. Furthermore, the temperature-dependency results in a smaller inter-quartile range of mE (quantified as the difference between $q_{25}$ and $q_{75}$ (upper and lower box bound in Fig. 7)) leading to more
accurate predictions of mE over all stations compared to the non-temperature-dependency variants. This trend is evident for all temperature classes, but its effect is most pronounced in the mean temperature classes. The distance between $q_{25}$ and $q_{75}$ is around 0.7 mm/5min without temperature-dependency and decreases to 0.3 mm/5 min with temperature-dependency.

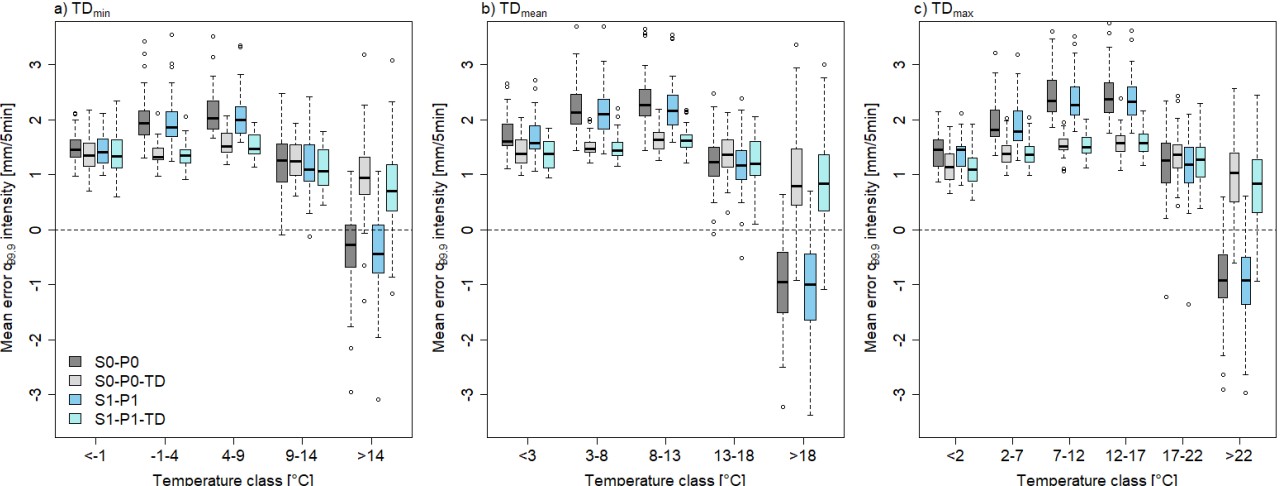

**Figure 7: Mean error of the temperature-dependent disaggregation on the observed $q_{99.9}$ rainfall intensity [mm/5 min] for TDmin**
**(a), TDmin (b) and TDmax (c) in the different temperature classes across all stations.**

In Fig. 5 the impact of temperature-dependency on the $q_{99.9}$ is shown in detail for the location A-E. Notably, the exponential behaviour over all temperature classes can be represented with the temperature-dependent disaggregation, which was not possible before. While for the highest temperature class (>18 °C), all model variants without temperature dependency show a great underestimation of $q_{99.9}$. S1-P1-TD leads to a better representation. A slight overestimation of $q_{99.9}$ for the highest
temperature class is identified in general, but also strong overestimations are possible (location C). In the lower temperature classes (≤18 °C) there are also overestimations of $q_{99.9}$. Furthermore, in addition to the $q_{99.9}$ rainfall intensity the impact of temperature-dependency was analysed on the event-based rainfall amount with T = 2 yrs (Fig. 8). Without temperature-

dependency, the median of rE for T = 2 yrs was found to be 40 % for the model variants. However, with the implementation of temperature dependent parameters, rE decreases to approximately 15 %. This improvement was observed across all temperature characteristics, and the differences among the temperature characteristics were negligible.

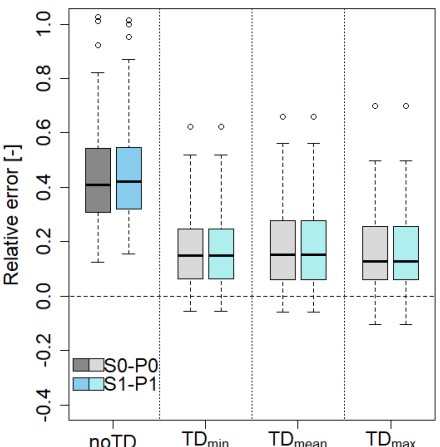

**Figure 8: Relative error of rainfall extreme values with D = 5 min and T = 2 yrs for the disaggregation without temperature (noTD) and temperature dependency (TDmin, TDmean and TDmax) across all stations.**

The aim of the temperature-dependent modification of the cascade model was to provide a physically inspired extension to the model to increase its applicability for future conditions. The temperature-dependent modification led to an improved representation of the rainfall intensity (mean and standard deviation) and slightly reduced wet spell amount.

Regarding the rainfall extreme values, the temperature-dependent modification had varied effects. Its introduction led to a reduction of mE of the $q_{99.9}$ rainfall intensity. The previous under- and overestimation of different quantiles were replaced by a smaller and invariable overestimation over all temperature classes. The notable advantage of the invariable deviation lies in its ease of interpretation, predictability and potential mitigation. These findings are particularly relevant for error analysis in model prediction. Also, the rE of the two-year return period was reduced for all stations, resulting in a better prediction of rainfall extreme values.

The introduction of temperature-dependency improves the cascade model, particularly with regard to extreme values. The minor difference in the results of the different temperature characteristics can be explained meteorologically. Similar rainfall events were observed in the temperature classes of each temperature type. The majority (>50 %) of days with a maximum temperature of >22 °C had a mean temperature of >18 °C and a minimum temperature of >14 °C (Tab. 8). Since similar rainfall time series in the temperature classes across the temperature characteristics led to comparable parameters of the temperature-dependent disaggregation model, the results were also similar. To analyse the difference between the temperature characteristics more precisely, smaller temperature class widths (e.g. 2°C) could be selected. However, this was not feasible due to the restricted time series length, which would have led to a small number of rainfall values in individual classes and larger uncertainty in of $q_{99.9}$ estimates.

**Table 8: Proportion of identical rainfall time steps [%] that can be found in each temperature characteristic (Tmin, Tmean and Tmax) for the highest temperature class (Tmin: >14 °C, Tmean: >18 °C and Tmax: >22 °C) for the locations A-E and mean across all 45 locations.**

| Temp. char. | Proportion of the same rainfall time steps [%] | | | | | |
|---|---|---|---|---|---|---|
| | A | B | C | D | E | ∅ 45 loc. |
| $T_{min}$ | 83 | 79 | 65 | 85 | 81 | 75 |
| $T_{mean}$ | 66 | 55 | 59 | 65 | 69 | 65 |
| $T_{max}$ | 50 | 34 | 41 | 43 | 56 | 52 |

## 4.3 Climate scenario disaggregation

Extreme rainfall events are more likely to occur in the summer when temperatures are higher (see Table 6). The expected changes of rainfall amount and mean temperature on wet days during the summer are shown in Fig. 9 for all locations. In general, the summer rainfall amount will change only slightly in both climate scenarios compared to C20. For the RCP 4.5 scenario, the rainfall amount increases by approximately two percent in most locations for the NTF and LTF. In contrast, for RCP 8.5 scenario the rainfall amount will decrease about two percent in the LTF, across the majority of locations.

The future mean temperature on wet days will increase by approximately 1.1–1.4 °C in the NTF under both, the RCP 4.5 and RCP 8.5 scenarios. The differences between the two climate scenarios are small in the NTF. However, for the LTF the mean temperature on wet days is expected to increase by about 1.6 °C compared to control period for RCP 4.5 scenario, and by approximately 3.4 °C for RCP 8.5 scenario. The difference between $q_{25}$ and $q_{75}$ of each boxplot is small, indicating similar future changes at each location. The increase in temperature underscores the importance of considering temperature-dependent parameters in the disaggregation of rainfall time series with the aim of extreme value analysis.

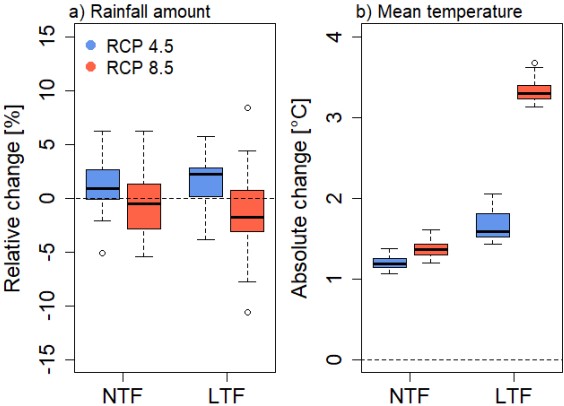

**Figure 9: Relative change [%] of rainfall amount (a) and absolute change [°C] of daily mean temperature on wet days (b) in the summer (April-September) between C20 (1971-2000) and the NTF (2021-2050) and the LTF (2071-2100) respectively, for RCP 4.5 and RCP 8.5 across all locations.**

To analyse the future change of rainfall extreme values, the daily rainfall time series from the climate scenarios were disaggregated to 5 min rainfall time series at each location, taking into account temperature dependency (S1-P1-TD).

Subsequently, rainfall extreme values with return periods T = 2 and T = 10 years were calculated following Eq. 3 for the rainfall duration of 5 min and 1 h. As shown in Fig. 10, in the NTF the differences between the two climate scenarios are small
(deviations of the medians <5 %) for both return periods and rainfall durations. The rainfall amount will increase by approx. 5–10 % in the NTF for both rainfall durations. However, the changes for RCP 8.5 is slightly higher than those for RCP 4.5. The future changes in the LTF exhibit considerable differences between both scenarios. For RCP 4.5 scenario, the rainfall amount is expected to increase by approximately 10 % for both return periods and rainfall durations. The changes compared to the NTF are relatively small for RCP 4.5 scenario. However, for T = 10 yrs and D = 1 h the $q_{75}$ of the boxplot is about 17.5
% indicating the highest increase for some locations for RCP 4.5. For a few locations a negative change (-0.5 to -2.5 %) can be identified.

For LTF for RCP 8.5 scenario an increase of about 20 % compared to the C20 period is identified, for some locations an increase of up to 50 %. This applies for both return periods and rainfall durations.

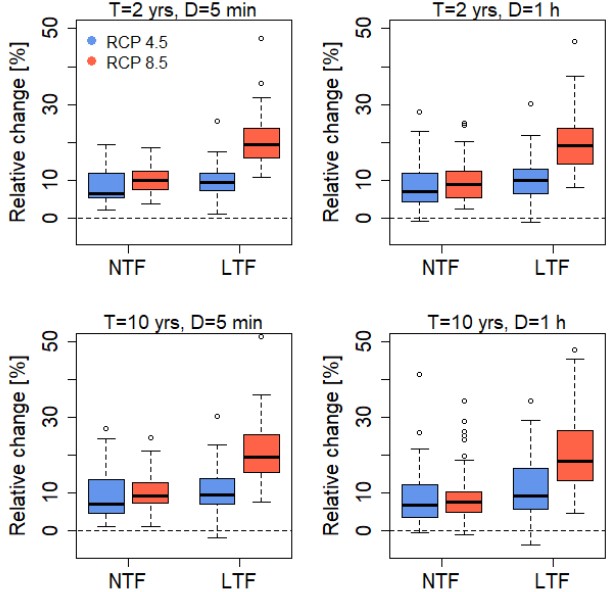

**Figure 10: Relative change of the rainfall amount for a rainfall duration (D) of 5 min and 1 h with a return period of two years (T = 2 yrs) and ten years (T = 10 yrs) between the control period C20 (1971-2000) and the near-term future NFT (2021-2050) and the long-term future LFT (2071-2100), respectively for RCP 4.5 and RCP 8.5 across all locations.**

To underline the importance of temperature-dependency, in Tab. 9 the relative changes of the future rainfall amounts for rainfall extremes with T = 2 yrs and T = 10 yrs are shown for the disaggregated rainfall time series with and without
temperature-dependency. A clear impact of the temperature-dependent disaggregation can be identified. Considering temperature-dependency the relative change is approximately doubled. The highest impact can be identified for location C for T = 10 yrs. Without temperature-dependency, the relative change is 12 %, whereas with temperature-dependency, it increases to 34 %. Conversely, the smallest impact can be identified for location D with a difference of 5 %.

**Table 9: Comparison of the relative change of the rainfall amount with the return period T = 2 years and T = 10 years for a rainfall duration of D = 1 h between the control period C20 (1971-2000) and the long-term future LFT (2071-2100) for RCP 8.5 for the disaggregation without temperature (S1-P1) and temperature dependency (S1-P1-TD) for the locations A-E and across all locations.**

| Locations | Relative change C20 - LTF [%] | | | |
| | T = 2 yrs | | T = 10 yrs | |
| | S1-P1 | S1-P1-TD | S1-P1 | S1-P1-TD |
|---|---|---|---|---|
| A | 14 | 23 | 15 | 24 |
| B | 14 | 24 | 12 | 26 |
| C | 12 | 28 | 12 | 34 |
| D | 9 | 13 | 8 | 13 |
| E | 15 | 26 | 16 | 26 |
| ⌀ 45 loc. | 12 | 21 | 13 | 22 |

The spatial distribution of the rainfall extreme event changes is shown in Fig. 11 for D=5 min and T=10 yrs (similar for T=2 yrs and D=1 h). In the northeast region of Germany, there is only a slight increase in the rainfall volume, ranging from >5 – <=15 % for T = 10. Conversely, the locations in the south exhibit the highest increase, exceeding >30 %. In the northern part and central Germany, the majority of locations experience an increase of 15 to 25 %. However, the identified spatial pattern is not homogenous, e.g., some locations show different changes of extreme values. To analyse the spatial differences between the locations further studies are required. One possible factor that can impact the relative change is the elevation of the location, which influences temperature.

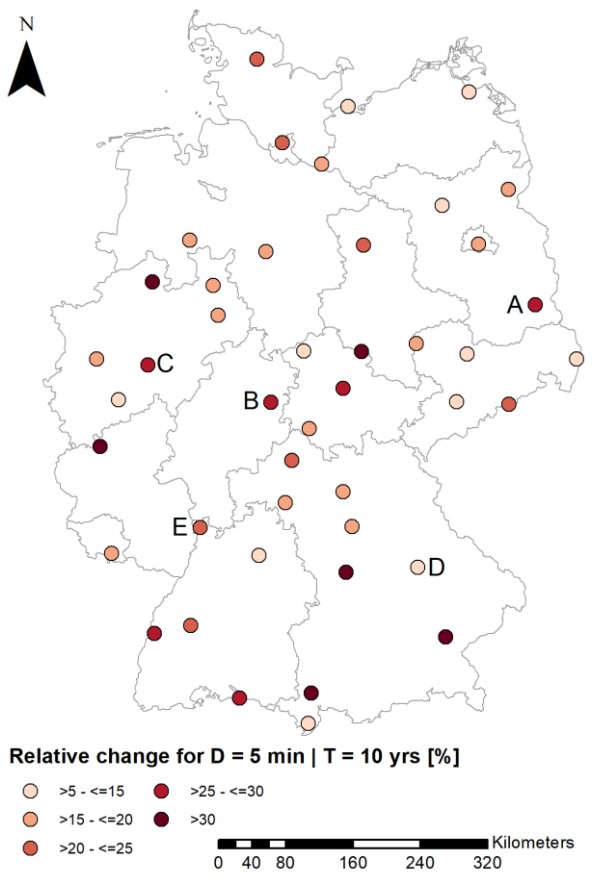

**Relative change for D = 5 min | T = 10 yrs [%]**

| | |
|---|---|
| ○ >5 - <=15 | ● >25 - <=30 |
| ○ >15 - <=20 | ● >30 |
| ● >20 - <=25 | |

Kilometers
0  40  80     160     240     320

**Figure 11: Relative change of rainfall extreme events for D = 5 min and T = 10 yrs between C20 (1971-2000) and the LTF (2071-2100) for RCP 8.5 at each location.**

In summary, based on the climate scenario data, no significant changes are expected in terms of seasonal rainfall amount in the summer in the future. However, temperatures are projected to increase, particularly for RCP 8.5 scenario. Therefore, it is crucial to take temperature-dependency into consideration in the disaggregation of rainfall time series for the analysis of rainfall extreme values.

In this study climate, scenario data was disaggregated from daily to 5 min time series, and the extreme values of the disaggregated time series were analysed. The results indicate that in the future, the rainfall volume of extreme events, based on a temporal resolution of 5 min, will increase by approximately 5–10 % in the NTF and 15–25 % in the LTF. Additionally, for certain locations, an increase of more than 30 % is observed.

Daily rainfall extreme values will increase at a rate of approx. 7 %/K. This increase aligns with the available water vapor, which increases with rising temperatures depending on the Clausius-Clapeyron relation (Seneviratne et al., 2021). For sub-daily rainfall extreme events a similar rate prevails, albeit with regional variations. For Europe, Lenderink and Meijgaard (2008) have identified an increase at a rate of 14 %/K, based on a coarse resolution RCM. Conversely, Hodnebrog et al. (2019) found out that sub-daily rainfall extremes do not increase with a rate above 7 %/K, caused by robust summer drying over large

parts of Europe. In the present study the sub-daily rainfall extreme events increase by 15-25 % in the LTF in RCP 8.5. This corresponds to an increase of 5-7 %/K on average across all locations. Notably, this value aligns to the rate from the previously mentioned studies. However, it should be noted that the increase is strongly dependent on the location, with locations showing an increase of 30-50% (8-14 %/K) but also stations with an increase of 10% (~3 %/K).

The difference between the RCP 4.5 and RCP 8.5 scenarios in the LTF is also evident in various studies and results mainly from the higher increase in temperature in the LTF in RCP 8.5 (~3.0°C) than in RCP 4.5 (~1.9 °C). In the NTF, the approximate increase of rainfall extreme values is for both scenarios 5 to 10%, with a temperature increase of 1.1 to 1.4°C, resulting in an increase of 5-7%/K.

Poschlod and Ludwig (2021) also analysed the future change in the return period of sub-daily rainfall extreme events,
identifying a change of 20-25 % for T = 10 and D = 1 h for the LTF in central Europe, which is comparable with the results of this study. Furthermore, the results are confirmed by a comparison with a convection-permitting climate model (see supplementary material S2 for details).

The key assumptions for the application of cascade models for the disaggregation of future climate model data is the stationary scaling behaviour of rainfall, which was empirically shown to be reasonable in the study area with additional data in the
supplementary material. Therefore, the parameter estimation is carried out data-driven and no calibration on future climate conditions is required. For this reason, the cascade model is the most promising method for statistical downscaling of climate model data to generate future sub-hourly rainfall time series.

A proof-of-concept of the temperature dependent disaggregation model is provided in the supplementary material with data not used in this study itself. An additional 5 min rainfall time series covering a long observation period (45 yrs) is split in two
periods, the temperature difference between both periods is derived and the changes of the extreme values are analysed for both periods for the observed and disaggregated time series. The results of this approach show that the temperature-dependent disaggregation model can reproduce an increase in rainfall extreme values induced by an increase in temperature. An application for climate scenario data affected by a temperature increase to analyse future changes in rainfall extreme values is therefore permissible.

**5 Summary and Conclusion**

The aim of this study is to analyse future rainfall extreme values in Germany on a sub-hourly time scale. For the disaggregation of daily rainfall time series of the climate scenarios a micro-canonical cascade model (Müller and Haberlandt, 2018) was refined. Modifications include introduction of temperature dependency and possibilities for parameter reduction. For the parameter reduction intra-event similarities and scale invariance were assessed with the following conclusions:

1. Parameter reduction based on intra-event similarities (P1) had negligible effects on the rainfall statistics of the disaggregated time series.
        2. Parameter reduction based on scale invariance has a stronger impact on rainfall statistics of the disaggregated time series. While the usage of two unbounded parameter sets (for the disaggregation levels S1, $\Delta t = 8$ h $\rightarrow$ 1 h and

$\Delta t$ =1 h → 7.5 min) resulted in only slight changes of the rainfall statistics, the usage of one bounded parameter set (S2, $\Delta t$ = 8 h→7.5 min) has a negative impact on the disaggregation performance (e.g. wet spell amount and $q_{99.9}$).

3. Overall, the best modification was the S1-P1 parameter reduction approach combining the scale invariance of rainfall properties (S1) and the intra-event similarities (P1) with 40 parameters instead of 144 without affecting the rainfall statistics of the disaggregated time series.

Temperature dependency was introduced to provide a physically inspired extension to the model to enhance its applicability for future conditions. Therefore, temperature classes with a class width of 5 °C were introduced, and the temperature dependency was analysed for the minimum temperature, mean temperature and maximum temperature. The findings are:

4. Sub-daily extreme rainfall events are temperature-dependent and predominantly occur at higher temperatures.
5. Introduction of temperature dependency improved the $q_{99.9}$ rainfall intensity and rainfall extreme values for T=2 yrs, while continuous and event-based rainfall statistics showed only slight changes.
6. There were only slight differences between the minimum temperature, mean temperature and maximum temperature.

The results of this study show that a temperature dependency of the cascade model parameters is relevant especially for the rainfall extreme events. Hence it is worth to analyse if the application of temperature-dependent cascade model parameters can be reduced to time steps with high rainfall intensities only.

Using the temperature-dependent disaggregation model, the daily climate scenario rainfall time series were disaggregated to 5 min resolution. The rainfall extreme values of the disaggregated time series were then analysed for the near-term future NTF (2021-2050) and long-term feature LTF (2071-2100) and compared with the control period C20 (1971-2000), leading to the following conclusions:

7. The rainfall amount of extreme events is projected to increase by 5-10 % for NTF and 15-25 % for LTF. However, the increase for RCP 8.5 scenario will be higher compared to the RCP 4.5, especially for LTF.
8. Taking temperature-dependency into consideration in the disaggregation of rainfall time series for the analysis of rainfall extreme values is crucial, as it leads to increases of >100 % of the changes of future rainfall extreme events (compared to the disaggregation without temperature-dependency).

Rainfall disaggregation proves to be a valuable tool for increasing the temporal resolution of climate scenario data. This approach allows for an analysis of finer temporal rainfall patterns and their impacts. The methodology outlined in this study, particularly the disaggregation process, is not limited to the specific climate model data used. It can be applied to other climate scenario data (e.g. CMIP6) and other temporal resolutions (e.g. 3 h). This transferability has the potential to enhance the accuracy of climate model impact studies.

In addition to the analysis of future projections of rainfall extreme values, the generated continuous 5 min rainfall time series offer significant potential for various applications, e.g. erosion studies, rainfall-runoff modelling, urban hydrological models und hydraulic models. The accuracy and effectiveness of these applications depend on the temporal resolution of the input time series, especially at small spatial scales. However, further studies are required to explore the temperature-rainfall extreme values relationship to take into account e.g. urban heat island effects at a finer spatial scale, which demands also disaggregation in space.

*Code and data availability*. The temperature data are accessible from the Climate Data Center web portal of the German Weather Service (https://opendata.dwd.de/climate_environment/CDC/observations_ germany/climate/, DWD, 2023). The YW-rainfall raster dataset is also accessible from Climate Data Center web portal of the German Weather Service (https://opendata.dwd.de/climate_environment/CDC/grids_germany/5_minutes/radolan/, DWD, 2023). The climate scenario data is available on request from the German Weather Service. The rainfall disaggregation program as well as the resampling 590 program are both written in Fortran and can be shared on request.

*Author contributions*. NE: conceptualisation, methodology, software, writing – original draft preparation. KS: writing - review & editing. HMT: supervision, conceptualisation, software, writing - review & editing.

*Competing interests*. At least one of the (co-)authors is a member of the editorial board of Natural Hazards and Earth System Science. The peer-review process was guided by an independent editor, and the authors have also no other competing interests to declare.

*Financial support.* This research has been supported by the LAWAMAD project, Thünen Institut, Germany.

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
