# Peer review of "Estimation of future rainfall extreme values by temperature-dependent disaggregation of climate model data"

_EGUsphere, 2023_

## Author Comment (AC3)

**General Reply**

We thank all three reviewers for their constructive comments on the manuscript, which helped to improve it significantly. A detailed reply to all comments is provided below. For a better overview, the reviewers' comments are shown in black, our replies in blue and proposed changes in the manuscript bold and blue.

We identify two primary points of the critique raised in the major comments of the three reviewers, which we would like to address first:

  i) Used climate scenario data: daily temporal and 0.11° spatial resolution does not represent the "state-of-science" and the impact of the (substantial) biases in rainfall from these RCMs is not assessed

The description of the climate model data used in this study was found to be inadequate and incorrect in certain sections, as highlighted in the referees' comments. We would like to start by providing more details regarding the climate model data we used and elucidating its appropriateness for our analysis. Specific modifications made to the manuscript in response to these concerns are listed in detail in the replies to the reviewers' comments below.

Firstly, we acknowledge the reviewers' valid point that the climate scenarios used in this study may not fully represent the current 'state-of-science.' The driver of the climate models used in our analysis are based on the Representative Concentration Pathway (RCP) emission scenarios. The 'state-of-science' climate models are driven by a combination of the Shared Socioeconomic Pathways (SSPs) and the RCPs, as introduced in the IPCC 6th Assessment Report. However, it's essential to note that for this new generation of climate simulations (CMIP6), regionalized and bias-adjusted climate model data for Germany are not available yet. Therefore, in this study, we used climate model data from the CMIP5 simulations. Specifically, the core ensemble of the German Weather Service (DWD) was used for the two emission scenarios RCP 8.5 and RCP 4.5. These datasets were selected for our analysis of future changes in rainfall extreme values. We acknowledge that the initial description of the DWD core ensemble in the manuscript have been insufficient and needs to be revised. Nevertheless, it is worth emphasizing the advantages of the DWD core ensemble and why we have used it.

The DWD core ensemble is a subset of the DWD reference ensemble consisting of six ensemble members. Importantly, the core ensemble retains the full climatic variability observed in the reference ensemble. The composition of Global Climate Models (GCM) and Regional Climate Models (RCM) is included in the manuscript. The DWD core ensemble is recommended by the DWD specifically for conducting impact analyses related to future climate changes. Its principal advantage lies in its spatial resolution, with a grid size of 5x5 km, tailored to the geographic extent of Germany. Additionally, the climate model data within the core ensemble underwent a bias adjustment, which significantly enhances its reliability for regional applications (Hänsel et al. 2020). Thus, for Germany, the DWD core ensemble represents the 'state-of-technic' in terms of regionalized and bias-adjusted climate model data. However, it's important to acknowledge one limitation of the DWD core ensemble, which lies in its daily temporal resolution.

The first referee has suggested the utilization of climate model data from EURO-CORDEX for our analysis. It is noteworthy that EURO-CORDEX data offers a temporal resolution of 3 hours, a spatial resolution of 0.11° (approximately 12.5x12.5 km) and does not include further regionalization or bias adjustment. The absence of bias adjustment represents a substantial limitation of this climate scenario data. Nonetheless, the higher temporal resolution holds distinct advantages for the disaggregation and analysis of extreme values, making it a compelling

option for future investigations. The same applies to the climate model data provided by Leduc et al. (2019), which was also proposed to use by the first referee. Regionalization and bias-adjustment for Germany are also missing in this data set. It is important to recognize that this study serves as a proof of concept, with the application of the disaggregation model on climate model data being just one facet of our research, alongside examining temperature dependencies and parameter reduction within the model. The transferability of our approach on different/upcoming climate model data, indicates its scientific importance for the analysis of sub-hourly rainfall extreme values. This aspect needs to be revisited and emphasized in the conclusion section.

ii) Poorly formulated research questions

The reviewers raised concerns about the poorly formulated research questions. We agree with their assessment and have rephrased them. Our intent is to ensure the research questions are more precise and accurately reflect our study's research focus.

*How can a cascade model be modified to improve the disaggregation results regarding rainfall extreme values with a minimum increase in model parameters?*

This research question pertains to the first segment of our study, which is dedicated to the analysis of parameter reduction and the influence of temperature-dependent cascade model parameters on the disaggregation results. Parameter reduction becomes especially important due to the increase of model parameters induced by the introduction of temperature dependency. To the author's knowledge a similar parameter reduction cannot be found in the existing literature. Although e.g., Bürger et al. (2019) introduced a temperature-dependency for a simpler cascade model, the resulting increase and possible reduction of model parameter were not investigated.

*How will rainfall extreme values with a temporal resolution of 5 min change in the future across Germany?*

This research question focusses on the future changes of rainfall extreme values with a temporal resolution of 5 min for 45 locations across Germany. To the authors' knowledge, such an investigation has not been conducted so far on these scales. Only for a coarser temporal resolution (10min) and for a smaller spatial extent (Emscher-Lippe region in Germany) Bürger et al. (2019) conducted a study that involved the disaggregation of rainfall time series. In our research, we shift the focus to a finer temporal resolution (5 minutes) and extent the spatial scope to 45 locations across Germany.

**Review by Benjamin Poschlod, Referee #1**

**General comment:**

The authors present a two-fold study, which 1) modifies an existing micro-canonical disaggregation method to include temperature classes as covariate, and 2) applies this disaggregation on climate projections for future rainfall. The study area extends over Germany, where 45 locations are presented. Overall, the manuscript is well structured and the figures are clear. The method is relevant for impact modellers. However, I have major concerns regarding the disaggregation setup (daily to 5-min) and the investigated climate model ensemble.

We thank Benjamin Poschlod for his useful and constructive comments and the time, he spent on the manuscript. A point-by-point reply is provided below.

**Major comments:**

The first part of the introduction (L29-63), which deals with climate modelling and scenarios, and section 2.2 have imprecise or uncommon vocabulary in several places (some are noted in the minor comments). This might confuse the reader and should be adapted.

Both parts (introduction (L29-63) and sections 2.2) need to be and have been revised and are hopefully now more precise (please see our replies to the minor comments).

In the introduction, the availability of sub-daily precipitation data from EURO-CORDEX is not correctly described. Here (https://doi.org/10.24381/cds.bc91edc3), 3-hourly data from many simulations are available. However, also the hourly resolution is often stored at the local model institute, but might be accessed only upon request (as e.g. by Berg et al., 2019: https://doi.org/10.5194/nhess-19-957-2019 ). At the same spatial resolution, the CRCM5-LE provides 50 members of hourly precipitation data (Leduc et al., 2019: https://doi.org/10.1175/JAMC-D-18-0021.1). Hence, I'd argue that the "main task" for rainfall disaggregation features the disaggregation from hourly to e.g. 5-min resolution. The relative error of the overall disaggregation (Tables 5 & 7) and of 2-year return levels (Figs. 5 & 7) could be possibly lowered by disaggregation from hourly to 5-min resolution instead of daily to 5-min. Hence, I strongly recommend to apply the disaggregation from 3-hourly or hourly to 5-min resolution.

The choice of the climate model ensemble ("DWD core ensemble") at daily temporal and 0.11° spatial resolution does not represent the "state-of-science". 0.11° simulations are available at 3-hourly and hourly resolution (see comment above). Furthermore, for the representation of convection and short-duration rainfall extremes, higher-resolution set-ups ("convection-permitting") are found to be beneficial (Coppola et al., 2021: https://doi.org/10.1007/s00382-018-4521-8; Purr et al., 2021: https://doi.org/10.1002/joc.7012).

We agree with the reviewers' comment in general. The description of the availability of sub-daily rainfall data from EURO-CORDEX in the manuscript was inadequate. This aspect has been revised to emphasize the existence of climate model data with sub-daily temporal resolutions. Consequently, our study is focused on analyzing rainfall extreme events with a temporal resolution of 5 minutes. Nevertheless, it's important to note that, currently, there are no climate model data available at this fine temporal resolution.

One notable point is that the feasibility of using 3-hour time series as input for the disaggregation, instead of daily values, represents a significant aspect that will be incorporated into the revised discussion section. In addition to parameter reduction and temperature dependency of the cascade parameters, our study serves as a proof of concept regarding the statistical downscaling of climate model data with a cascade model to a temporal resolution of 5 min. Furthermore, the approach holds substantial promise in terms of transferability to other climate scenario data. In the revised Summary and Conclusion section, we have highlighted the potential to enhance the study results by utilizing 3-hour or 1-hour climate data time series as input for the disaggregation.

We decided against using the climate model data from EURO-CORDEX in our study. Instead, we decided for the DWD core ensemble recommended by the DWD for future climate change impact studies in Germany due to distinct advantages, including its regionalization at a 5x5 km raster resolution and its bias adjustment (Hänsel et al. 2020).

The reviewer argues that the "main task" of rainfall disaggregation is the disaggregation from 1 h to sub-hourly temporal resolution. We respectfully disagree with this perspective. Daily rainfall observations are widespread

around the world and cover much longer periods compared to sub-daily or sub-hourly rainfall observations. To take advantage of these extensive records, the disaggregation from daily to finer temporal resolutions is of great importance and finds applications in various fields and studies (e.g. Acharya et al. 2022, Breinl & Di Baldassarre 2019, Guan et al. 2023). So, although the second part of the manuscript focusses on climate change impact on rainfall extreme values, the parameter reduction and introduction of temperature-dependency remains an important step forward for the disaggregation of observed rainfall data, especially for sparsely observed regions.

The possible post-processing (downscaling? bias-adjustment?) of the climate model data is totally unclear. Neither the article nor the provided reference (Delalane, 2021) does provide the necessary information. Convection-permitting simulations would be available for Germany from the German Weather Service (journal article: Rybka et al., 2022, https://doi.org/10.1127/metz/2022/1147; data: https://dx.doi.org/10.5676/DWD/HOKLISIM_V2022.01; https://esgf.dwd.de/projects/dwd-cps/cps-hist-v2022-01; https://esgf.dwd.de/projects/dwd-cps/cps-scen-v2022-01 ).

We thank the reviewer for pointing at the missing references on the post-processing of the climate model data. The DWD core ensemble used in this study has undergone bias-adjustment and downscaling (Hänsel et al. 2020). Both processes were carried out by the Federal Ministry for Digital and Transport - Network of Experts Topic 1 (BMVI-Network of Experts Topic 1) and were not within the scope of this study. Nevertheless, it is essential to underscore that these downscaling and bias-adjustment procedures are pivotal when working with climate model data, offering a distinct advantage of the DWD core ensemble over climate model data from EURO-CORDEX. References on post-processing methods for climate model data should be incorporated into the manuscript, and a revision of section 2.2 will be undertaken to address this issue.

In response to the availability of convection-permitting models, we acknowledge that such simulations were not a viable option for our study due to the currently limited selection of one available convection-permitting RCM. This contradicts with the ensemble-approach in climate change impact studies, to account for climate variations within the climate models. Therefore, we have analyzed in our study multiple ensemble members for each RCP scenario.

The discussion section only discusses projected increases of sub-daily extreme rainfall and its temperature scaling compared to three other studies. However, the disaggregation procedure and its performance compared to other approaches is not discussed. Pui et al. (2012) follow that the method of fragments (MoF) outperforms a micro-canonical cascade model. The systematic underestimation of the wet spell duration (Table 5) is also found by Pui et al. for the cascade model, whereas MoF can reproduce this rainfall characteristic (see also Poschlod et al., 2018: https://doi.org/10.1175/JHM-D-18-0132.1 for a comparison to convection-permitting climate model performance). The authors should at least discuss these drawbacks and elaborate on the advantages of a cascade model versus the MoF or other disaggregation approaches (e.g. Zhao et al., 2021: https://doi.org/10.1016/j.jhydrol.2021.126461 ).

We thank the reviewer for his general questioning of the choice of the rainfall generator. We have mentioned other rainfall generators in the manuscript (L67: "Well-known disaggregation methods are the method-of-fragments (MoF, Westra et al. 2012), Bartlett-Lewis rectangular pulse model (BLRP, Koutsoyiannis and Onof, 2001, Onof and Wang, 2020) and cascade models (Molnar and Burlando 2005, Paschalis et al. 2012, Müller and Haberlandt, 2018). An overview of different rainfall disaggregation methods is provided by Pui et al. (2012)"), but went for

the cascade model because (in our opinion) it is the rainfall generator with the lowest restrictions for future climate data applications.

The main restriction of the MoF is that only observed patterns of rainfall (fragments) can be reproduced, which disables the representation of a high-convective event in time if it is not included in the observations. The temperature-dependency introduced in this study uses temperature classes, which limits the amount of fragments for the highest temperatures and hence will lead to similar diurnal cycles for these temperatures. The cascade model is based on probabilities, which allow numerous different diurnal cycles based on the self-similarity of the temporal scales only. This enables rainfall extreme value changes of different magnitudes on different temporal scales (Derx et al., 2023).

The main restriction of the pulse models is the assumption if model parameter distributions (pulse length, pulse rainfall amount, interarrival time of pulses,…) will change, and if yes, how. The probabilities used in the cascade model relate changes on a finer scale to changes on the coarser scale (which is provided on the daily scale by the climate scenario). So, the only assumption for the cascade model is that the physical rainfall process can be described in the future with the same statistics as done for the past - so that scaling behavior remains stationary.

The possible extension to the rainfall generator choice in the manuscript would be:

**"Since for the method-of-fragments the amount of fragments is limited by the observation length (critical especially in combination with temperature-dependency), and for the Bartlett-Lewis rectangular pulse model assumptions about distribution function of pulse characteristics have to be made, the cascade model was chosen for this study. The 'only' assumption for the application of cascade models for the disaggregation of future climate model data is that the scaling behaviour of rainfall remains stationary, which is not questioned to the authors knowledge."**

Regarding the wet spell duration: The wet spell is not systematically underestimated by the cascade model, also overestimations occur (Derx et al., 2023). However, the wet spell duration strongly depends on the threshold applied for considering a time step as wet or dry (e.g. Müller and Haberlandt, 2018, Fig. 5 vs. Fig. 6). As shown by Pidoto et al. (2022, supplementary material Fig. S1, provided below) the variation of the threshold influences the resulting statistics significantly. The concern raised by reviewer 3 with the dry duration considered for event/spell independency also has a crucial impact on this rainfall characteristic. Also, it remains unclear for the authors why the wet spell duration as a characteristic of all (small, moderate and high rainfall intensities) should be used as decision criteria while the focus of the submitted manuscript is on extreme values).

[Figure]

Figure S1: Sensitivity of erosive event characteristics and R to measuring resolution. Thresholds on the x-axis were used to replace smaller rainfall amounts by 0 mm for each 5 min time step before estimating erosive event characteristics. Outliers were excluded due to image clarity. Results are based on all 5min stations in the study area (figure is from Pidoto et al., 2022, supplementary material)

Furthermore, general uncertainties of the whole workflow need to be discussed: 1) The applied climate models have model biases. They parameterize convective processes. 2) What happens for temperature values under e.g. RCP8.5 LTF, which are outside the range of observed reference temperatures? 3) more generally: this empirical disaggregation method is calibrated on reference climate, but applied on strongly altered climate. This should be acknowledged from my perspective.

We concur with the reviewer's observation that certain aspects of our workflow require more extensive discussion:

1) It's important to note that the climate model data used in our study have undergone bias adjustment, effectively minimizing the impact of bias.

2) Temperature values outside the range of observed reference temperatures are assigned to the lowest and highest temperature class, which has no lower and upper limits, respectively. However, it is worth to note that such temperature values were not identified in the RCP LTF.

3) Regarding the disaggregation model, the model parameters were estimated from observations, under the assumption that the physical process behind rainfall and its statistical characterization will remain identical under future conditions. This assumption is a key aspect for the choice of this rainfall generator. The parameter estimation is carried out data-driven, so no calibration/iterative fitting or similar takes place.

**Minor comments:**

L30: "predicted" --> "projected"

The word "predicted" was changed to "projected".

L35: How does disaggregation of daily rainfall relate to question 9: 'How do flood-rich and drought-rich periods arise, are they changing, and if so why?'? Can you elaborate on that?

The disaggregation of daily rainfall is directly relevant to addressing question 9: 'How do flood-rich and drought-rich periods arise, are they changing, and if so why?' This relevance is underscored by the fact that rainfall time series with sub-hourly temporal resolution play a pivotal role in understanding hydrological processes, particularly during flood-rich periods, on the micro- to meso-scale.

However, it's essential to acknowledge that rainfall time series with sub-hourly resolution are not consistently available in space and time. To address this limitation, the disaggregation of daily rainfall time series serves as a viable solution. Furthermore, this process is of particular importance assessing changes in flood-rich periods at the micro- to meso-scale in the context of future climate scenarios. By disaggregating future daily rainfall time series to sub-hourly temporal resolutions, we enable a more comprehensive analysis of these changes. We would like to add:

**"In particular, sub-hourly rainfall extreme events are important to analyze the number of pluvial floods, which are relevant for mesoscale and finer, rather than fluvial floods, which are relevant for meso- and macro-scale."**

L39: RCP are emission scenarios not "climate scenarios".

The reviewer is correct the RCPs are emission scenarios and not "climate scenarios".

L40: IPCC 6th AR and the "state-of-science" in CMIP6 applied scenarios, which combine Shared Socioeconomic Pathways (SSPs) and the RCPs (Riahi et al., 2017: https://doi.org/10.1016/j.gloenvcha.2016.05.009). EURO-CORDEX is driven by CMIP5. Please clarify in this paragraph.

We agree with the reviewer regarding the application of 'state-of-science' scenarios in CMIP6, which integrate the Shared Socioeconomic Pathways (SSPs) and Representative Concentration Pathways (RCPs). Nonetheless, it's important to differentiate that while CMIP6 scenarios represent the 'state-of-science' on a broader level, they are notably limited when it comes to regionalized and bias-adjusted data at the moment. The term 'state-of-science' in our context was specifically linked to the availability of regionalized and bias-adjusted climate model data. This kind of data is predominantly accessible through simulations in CMIP5, which are driven only by the RCPs so far. We will make the following changes:

**"In the sixth assessment report the IPCC established the "Shared Socioeconomic Pathways" (SSP) – scenarios representing a range of social-economic trajectories into the future. The Coupled Model Intercomparison Project (CMIP) coordinates the climate model simulations globally and is responsible for the new climate model generation of the sixth phase (CMIP6) in which the RCP and SSP scenarios are combined. However, for the CMIP6 there are no bias-adjusted and regionalized climate model data available for Germany so far, but for CMIP5 these climate model data exist. So CMIP5 is considered as 'state-of-technic' climate simulations for Germany."**

L41: Please rephrase: GCMs do not model RCP scenarios, as the emissions are provided as part of the external forcing to the GCMs.

The sentence should be rephrased to:

**"In CMIP5 the RCP scenarios are provided as part of the external forcing to the Global Climate Models (GCMs), which simulate climate projections on the global scale."**

L91-104: Here, you already introduce quite specific features of the cascade model in order to discuss the parameter reduction. However, the reader does not know about the parameters beforehand. I'd recommend to shift this paragraph to Sect. 3.2.

We propose to relocate L102-104 to section 3.2 and added:

**"A detailed description of the intra-event similarities can be found in Sec. 3.2 to avoid technical details in this section."**

This decision aligns with the structure of our paper. In the paragraph, we provide the literature background for both approaches for parameter reduction, with the first part (L91-99) focusing on the scale dependency of model parameters. This portion does not delve deeply into specific model properties and can be comprehended by readers without in-depth knowledge of the Cascade model.

L127: Updated reference to Köppen-Geiger climates is available from Beck et al., 2018: https://doi.org/10.1038/sdata.2018.214

The reference has been updated to: **"Beck et al., 2018"**

L132: The sentence is not well connected; first you describe the overall climatology and then jump to availability of temperature data. I'd recommend to reorganize 2.1 into paragraphs on 1) climatology of Germany, 2) station data, 3) radar data.

The section 2.1 will be reorganized to the structure recommended by the reviewer.

L139 / Fig. 1: Please provide a readable legend for elevation.

The legend of Fig. 1 will be updated.

L162: The spatial resolution is described in the abstract (5km, L17), but misses here. How is the spatial downscaling from 0.11° to 5km carried out? Are the data bias-adjusted? Is drizzle considered/removed?

The spatial downscaling from 0.11° to 5km of the DWD core ensemble was conducted by the Federal Ministry for Digital and Transport - Network of Experts Topic 1 (BMVI-Network of Experts Topic 1). Prior to this spatial downscaling process, the climate projections underwent bias adjustment, a procedure also administered by BMVI-Network of Experts Topic 1 (Hänsel et al. 2020). For the bias adjustment of rainfall data, the quantile mapping method was employed. Quantile mapping methods serve to align the frequency distributions of projected data, which are prone to errors, with those derived from real observations through an average mapping rule. This adjustment encompasses parameters like the mean, standard deviation, individual quantiles, and even extrema. It is important to note that this methodology operates under the assumption of stationary frequency distributions. Nevertheless, it is recognized that this assumption may not hold true for all climate parameters in a changing climate.

The spatial downscaling of the bias-adjusted results from regional climate models involved the use of multiple linear regression, where typical distribution patterns of the respective climate variables served as predictors. This process was followed by the interpolation of the regression residuals. The underlying assumption was that regional climate models accurately reproduce the coarse-scale patterns of the climate variables being regionalized. In the regionalization process, the fine-scale structures of these climate variables are incorporated using the typical patterns derived from high-resolution reference data.

It is worth noting that we have thoroughly revised the entirety of section 2.2 to provide a more detailed description of the DWD core ensemble, with a particular focus on spatial downscaling and bias adjustment methods. However, both processes were not in the scope of this study, hence we will add the reference for it at the respective locations.

L174ff.: The explanation of the cascade model should be rearranged. A general explanation of the working principle of a cascade model is missing. In L183, you mention the model parameters are scale-dependent without having introduced the model parameters (only the parameter b; parameters are fully introduced in L204 for the first time). The paragraph should be revised and understandable by a reader without pre-knowledge about the cascade model.

Section 3.1 has been revised and some paragraphs have been rearranged. At the start of the section a short explanation of the working principle of a cascade model should be added.

**"A cascade model is used to increase the temporal resolution of a rainfall time series by distributing a coarse rainfall time step into finer time steps. This process is known as disaggregation. The number of resulting wet time steps and their rainfall amount depends on the cascade generator. However, in each disaggregation step the rainfall amount is exactly conserved. The cascade model parameters are estimated from observed time series."**

L209: How sensitive is the resulting disaggregation to the choice of the volume class threshold (q=0.998)? Are there any "jumps" in the resulting rainfall frequency/intensity around the q=0.998? I would like to see a figure of resulting sorted rainfall intensities of different durations above q=0.99.

The volume class threshold of q=0.998 is only applied for first disaggregation step with b=3 (24h->8h). For b=2, the mean rainfall volume is used as the threshold.

The choice of the threshold value, q=0.998, is motivated by the fact that only a limited number of daily rainfall values surpass this quantile and are subsequently distributed across a single or two 8-hour intervals. This specific threshold value leads to significant distinctions in the parameters of the volume classes, as demonstrated in Müller (2016). The threshold is identified with the same method for other study regions as well (e.g. Switzerland, Ghana), although not mentioned in the respective manuscripts.

For q > 0.99 there are no discernible 'jumps' within the data (Fig.below).

[Figure]

Figure: Rainfall intensity for various rainfall durations and quantiles (q > 0.99) at location A.

L232: Can you additionally provide a measure of the intra-event similarity for all 45 locations?

One approach to assess intra-event similarity across all 45 locations involves measuring the change between the parameters for both similarities (Fig. S3). A smaller change indicates a higher degree of parameter similarity. For

all disaggregation steps and both volume classes the absolute change are low (0 – 2 %). We propose to add this figure with the following text in the manuscript:

**"In Fig. 4 the intra-event similarity for all 45 locations is showing only minimal changes (0-2 %) between the parameters for both similarities underlying our assumption. For similarity 1 the differences between $P_{starting}(0/1)$ and $P_{ending}(1/0)$ in $V_1$ are slightly higher compared to $V_2$. Contrary, in similarity 2 changes between the parameters are slightly higher in $V_2$. However, the difference is negligible and it can be assumed that the parameters are almost the same at all locations."**

[Figure]

Figure: Absolute difference [%] of the cascade model parameters from $P_{starting}(0/1)$ to $P_{ending}(1/0)$ (Similarity 1) and from $P_{starting}(1/0)$ to $P_{ending}(0/1)$ (Similarity 2) in both volume classes across all 45 stations and disaggregation steps.

L238/Tab 3: V index seems not correct.

The V index has been corrected.

L252ff: As the temperature-dependent cascade model performs better only for rainfall extremes (see Tab. 7, Fig. 6 and 7), the dependency could only be introduced for the volume class V2? In Section 3.3, the temperature dependency is estimated for temperature classes at 5°C steps, for all rainfall intensities > 0 mm / 5 min. However, the temperature dependency is expected to be different for different rainfall generating mechanisms (stratiform vs. convective), which are associated with different rainfall intensities.

We thank the reviewer for this suggestion, which would be a new kind of parameter reduction of the temperature-dependent cascade model. Especially for days with rainfall volumes close to the threshold applied for the volume classes this suggestion could lead to an alternate and hence inconsistent usage of temperature-dependent and temperature-independent parameters in different disaggregation steps, e.g. temperature-dependent model parameters for $\Delta t=\{8h{\rightarrow}4h, 2h{\rightarrow}1h{\rightarrow}30min\}$ and temperature-independent model parameters for $\Delta t=\{4h{\rightarrow}2h, 30min{\rightarrow}15min\}$. An inconsistent parameter usage in different disaggregation steps questions the self-similarity of rainfall on neighboring scales, which is the basic principle of the cascade model. A decision for temperature-(in)dependent model parameters only on the coarse starting scale (here daily) accompanies with other issues (sub-daily high-intense events will be ignored).

So, an implementation of the suggested parameter reduction requires additional scale-dependent investigations, especially regarding a possible violation of the cascade model principles. This would be a rather theoretical study,

which can't be implemented in the current manuscript. So, although the suggestion sounds interesting and promising, we can add it as outlook only, unfortunately:

**"The results of this study show that a temperature dependency of the cascade model parameters is relevant especially for the rainfall extreme events. Hence it is worth to analyze if the application of temperature-dependent cascade model parameters can be reduced to time steps with high rainfall intensities only."**

L270: What are the advantages of the DWA approach over the well-established extreme value theory approaches, such as peak-over-threshold sampling and Generalized Pareto distribution fit (Davison and Smith, 1990: Davison, A. C. and Smith, R. L.: Models for exceedances over high thresholds, J. Roy. Stat. Soc., 52, 393–442, 1990)?

The DWA approach is a peak-over-threshold approach (described in L269), which is complied with the extreme value theory. Instead of the GPD the plotting position formula is used as recommended in the national guidelines for the design and planning of drainage systems, such as DIN 1986. These guidelines rely on calculated rainfall extreme values, as outlined in the DWA-A 531 guideline.

Given the relevance of our study to rainfall extreme values in Germany, it was a logical choice to incorporate the DWA approach into our research. However, the usage of a different distribution function as GPD could only lead to a systematic deviation of the extreme values of certain return periods (e.g. rainfall intensities for T=10yrs could be in general higher or lower), so the results of the study (change from C20 to NTF or LTF) remain unaffected.

L279: Rainfall characteristics should be already introduced and defined in Section 3.4. How are wet spell duration, dry spell duration, and wet spell amount defined?

Wet spell duration, dry spell duration, and wet spell amount are first mentioned in section 2.1 L145. However, the definition of the individual rainfall characteristics can be a bit more detailed:

**"Following the event definition of Dunkerley (2008) a rainfall event is defined as a rainfall period enclosed by at least one dry time step. A dry time step refers to a rainfall intensity of 0 mm /5 min. The wet spell duration represents the duration of a rainfall event enclosed by two dry time steps. The wet spell amount is the sum of rainfall occurred during the wet spell. Dry spell duration is the duration of a dry period enclosed by wet time steps."**

L359: "scatter" means variance or inter-quartile range?

In this context scatter refers to the difference between the $q_{25}$ and $q_{75}$ and thus describes the inter-quartile range. To be more precise scatter was changed to inter-quartile range:

**"Furthermore, the temperature-dependency results in a smaller inter-quartile range of mE …"**

L470 / L604: The article in the references (L604: https://doi.org/10.5194/essd-13-983-2021) describes the evaluation of sub-daily extreme precipitation compared to observational products, whereas the article in the text (L470: https://doi.org/10.1088/1748-9326/ac0849) investigates future projections and temperature scaling of extreme rainfall.

The reviewer is correct. The article in the reference is wrong and has been corrected:

**"Poschlod, B. and Ludwig, R.: Internal variability and temperature scaling of future sub-daily rainfall return levels over Europe, Environ. Res. Lett., 16, 64097, https://doi.org/10.1088/1748-9326/ac0849, 2021."**

References:

Acharya, S. C., Nathan, R., Wang, Q. J., and Su, C.-H.: Temporal disaggregation of daily rainfall measurements using regional reanalysis for hydrological applications, Journal of Hydrology, 610, 127867, https://doi.org/10.1016/j.jhydrol.2022.127867, 2022.

Breinl, K. and Di Baldassarre, G.: Space-time disaggregation of precipitation and temperature across different climates and spatial scales, Journal of Hydrology: Regional Studies, 21, 126–146, https://doi.org/10.1016/j.ejrh.2018.12.002, 2019.

Derx, J., Müller-Thomy, H., Kılıç, H. S., Cervero-Arago, S., Linke, R., Lindner, G., Walochnik, J., Sommer, R., Komma, J., Farnleitner, A. H., and Blaschke, A. P.: A probabilistic-deterministic approach for assessing climate change effects on infection risks downstream of sewage emissions from CSOs, Water Research, 120746, https://doi.org/10.1016/j.watres.2023.120746, 2023.

Guan, X., Nissen, K., Nguyen, V. D., Merz, B., Winter, B., and Vorogushyn, S.: Multisite temporal rainfall disaggregation using methods of fragments conditioned on circulation patterns, Journal of Hydrology, 621, 129640, https://doi.org/10.1016/j.jhydrol.2023.129640, 2023.

Müller, H: Niederschlagsdisaagregation für hydrologische Modellierung, Heft 101, Institut für Wasserwirtschaft, Hydrologie und landwirtschaftlichen Wasserbau, 2016.

Hänsel, S., Brendel, C., Fleischer, C., Ganske, A., Haller, M., Helms, M., Jensen, C., Jochumsen, K., Möller, J., Krähenmann, S., Nilson, E., Rauthe, M., Rasquin, C., Rudolph, E., Schade, N., Stanley, K., Wachler, B., Deutschländer, T., Tinz, B., Walter, A., Winkel, N., Krahe, P., and Höpp, S.: Vereinbarungen des Themenfeldes 1 im BMVI-Expertennetzwerk zur Analyse von klimawandelbedingten Änderungen in Atmosphäre und Hydrosphäre, https:// https://doi.bafg.de/BfG/2020/ExpNHS2020.2020.01.pdf, 2020.

Pidoto, R., Bezak, N., Müller-Thomy, H., Shehu, B., Callau-Beyer, A. C., Zabret, K., and Haberlandt, U.: Comparison of rainfall generators with regionalisation for the estimation of rainfall erosivity at ungauged sites, Earth Surf. Dynam., 10, 851–863, https://doi.org/10.5194/esurf-10-851-2022, 2022.

**Review by anonymous Referee #2**

**Review of 'Estimation of future rainfall extreme values by temperature-dependent disaggregation of climate model data' by Ebers et al.**

The study analyzes a set of 45 sub-hourly rainfall stations, uniformly distributed over Germany, with respect to the occurrence of rainfall extremes. Future projections of corresponding quantities are obtained from an existing micro-canonical cascade model (Müller and Haberlandt 2018) that is extended to include temperature as a covariate. The resulting model parameters are reduced in two directions, one by assuming scaling behavior across the relevant time scales, and one by assuming certain intra-event symmetries; this reduces the number of parameters from 144 to 40.

The so optimized disaggregation is then applied to 6 GCM/RCM model simulations, each driven by a modest (RCP4.5) and by a pessimistic (RCP8.5) emission scenario.

The main results are:

– disaggregation performance with respect to key quantities (e.g. intensity) is improved by using T dependency

– in the near future, changes in the intensity of extreme (2y) events are moderate

– in the long term, especially for RCP8.5, changes are +12% without and +21% with T dependency.

It is also reported, but not discussed much, that core statistics such as wet spell duration or rainfall intensity deviate considerably for the disaggregation model, with relative errors of -22% and 23%, respectively.

We thank the reviewer for the effort and the time spend on this manuscript. His/her major concerns are the research questions of the study and the climate model data used in the study. Both concerns are pointed out in the general reply above. A point-by-point reply can be found below.

**Major comments:**

As they are posed, the three research questions of the study

i) Is there a temperature-dependency of sub-daily rainfall extreme values?

ii) How can the temperature-dependency be integrated in the cascade model parameters for temporal rainfall disaggregation?

iii) How will rainfall extreme values change in the future?

are not new. iii) is a core topic of IPCC since more than two decades, i) belongs to the folklore at least since the seminal Lenderink and Meijgaard (2008) paper, and the more specific point ii) has already been addressed e.g. by Bürger et al. (2019). To become publishable, the study evidently needs to re-formulate its goals decisively. One options is to dive more deeply into the strong biases (see above) and their dependency on parameterization. It is not clear, for example, why one should have parameter reduction at all (apart of course from the general validity of Occam's razor). Another is to emphasize more clearly the regional structure (the 45 stations) of the main results.

At the suggestion of the reviewer, we have adapted the research question of the study. The revised research question are more precise and are intended to better represent the research focus of this study.

**"i) How can a cascade model be modified to improve the disaggregation results regarding rainfall extreme values with a minimum increase in model parameters?"**

**"ii) How will rainfall extreme values with a temporal resolution of 5 min change in the future across Germany?"**

Another weakness is the uncritical and un-adjusted use of climate models, whose biases confound the disaggregation bias. What are the biases? Which models have which bias? And where? On this background, are relative projections (e.g. +21% intensity) still reliable and have added value?

The climate model data used in this study has been bias-adjusted. However, this aspect was not explicitly detailed within the paper, as it was not part of our work. The regionalization and bias adjustment of the climate model data was conducted by the Federal Ministry for Digital and Transport - Network of Experts Topic 1 (BMVI-Network of Experts Topic 1). In section 2.2, we have revised the description of the climate model data to provide a more

comprehensive account, including a method description for the regionalization of 0.11° to 5 km raster and the bias adjustment of both temperature and rainfall data.

Furthermore, it's crucial to recognize that the parameters of the cascade model were estimated from observed values. In the context of disaggregating future values, the only assumption for the cascade model is that the physical rainfall process can be described in the future with the same statistics as done for the past - so that scaling behavior remains stationary.

Although a bias-adjustment was carried out, there is a wide range between the climate models. Therefore, absolute values resulting from the individual climate ensembles are not provided and only the relative change is analyzed, which enables comparability between the climate models.

Specific comments:

l. 40: developments --> evolutions

Rephrased to **"evolutions"**.

l. 40: but IPCC AR6 uses SSPs

We agree with the reviewer that the IPCC AR6 employs Shared Socioeconomic Pathways (SSPs) scenarios. However, it's noteworthy that simulations for the SSPs are limited in terms of regionalization and bias adjustment so far. Consequently, for our study, we decided to use simulations from CMIP5, which are driven solely by the Representative Concentration Pathways (RCPs). It's important to emphasize that CMIP5 simulations exist as regionalized and bias-adjusted data for Germany.

We have made revisions to the relevant section to ensure clarity and alignment with our approach:

**"In the sixth assessment report the IPCC established the "Shared Socioeconomic Pathways" (SSP) – scenarios representing a range of social-economic trajectories into the future. The Coupled Model Intercomparison Project (CMIP) coordinates the climate model simulations globally and is responsible for the new climate model generation of the sixth phase (CMIP6) in which the RCP and SSP scenarios are combined. However, for the CMIP6 there are no bias-adjusted and regionalized climate model data available for Germany so far, but for CMIP5 these climate model data exist. So CMIP5 is considered as 'state-of-technic' climate simulations for Germany."**

l. 76-78: I don't understand this argument.

The cascade model parameters relate the distribution of rainfall amounts on finer time steps to the rainfall amount on the coarser time step. These parameters describe only probabilities for splitting possibilities of the total rainfall amount. They do not prescribe the precise values of wet spell duration or amount as in other rainfall generators at finer scale. Consequently, we assume that the cascade parameters can be applied for the disaggregation of future time series. This line will be rephrased to:

**"Since for the method-of-fragments the amount of fragments is limited by the observation length (critical especially in combination with temperature-dependency), and for the Bartlett-Lewis rectangular pulse model assumptions about distribution function of pulse characteristics have to be made, the cascade model was chosen for this study. The 'only' assumption for the application of cascade models for the**

**disaggregation of future climate model data is that the scaling behaviour of rainfall remains stationary, which is not questioned to the authors knowledge.”**

l. 82: Kelvin units are "K" not "k".

The typo was corrected to "K".

l. 94: note that already Olsson (1998) uses parameter reduction (by averaging over several levels).

The sentence will be revised to:

**"Olsson (1998) tested before an averaged parameter set estimated from coarser resolution (17h aggregated up to 5.7 days) for the disaggregation of time steps from ~17h to ~1h and found only a minor worsening of the disaggregation results. Olsson (1998) also showed that parameters estimated from time series with a temporal resolution <1 h differ from parameters estimated from a time series with a coarser temporal resolution."**

l. 108: All central questions have been addressed in previous work already:   i) has populated the scientific debate at least since the seminal paper by Lenderink    and Meijgaard (2008) and does not really fit as a core research question.

  ii) has been thoroughly addressed by the Bürger et al. papers.

  iii) is probably one of the most addressed questions in climate research of the past 3  decades or so.

Please see our general reply at the beginning of this document and the replies to the major comments.

l. 148: It is not clear to me why you used the YW data at all, and not just stick to the station data?

We aim to disaggregate climate model data with a spatial resolution of 5x5 km, and the cascade model parameters are estimated from observed data. To ensure spatial consistence for these parameters, we also estimated them using a 5x5 km raster. It's essential to note that rainfall observations at a 5x5 km raster are accessible through the aggregation of the 1x1 km YW data.

Rainfall statistics at a single rain gauge/point can differ significantly from rainfall statistics at a 5x5 km raster cell, particularly concerning rainfall extreme events (which is the main reason for the existence of areal reduction factors (ARFs).

l. 159: 'climate scenario' and 'climate emission' inadequate

The reviewer is correct, the RCPs are emission scenarios and not "climate scenarios". It has been corrected.

l. 164: there is a big resolution gap between the 50km/12.5km resolution of EURO-CORDEX and the 5km of the DWD. Please explain!

The spatial downscaling of the DWD core ensemble, from 0.11° to 5 km, was executed by the Federal Ministry for Digital and Transport - Network of Experts Topic 1 (BMVI- Network of Experts Topic 1) employing the multiple linear regression technique (Hänsel et al. 2020). In response to the reviewer's comment, we have revised section 2.2, providing a more comprehensive description of the climate model data, including an in-depth account of the spatial downscaling method and the bias adjustment process.

l. 172: Since for the main part a branching number of b=2 is used, it is unclear how the cascade model is different from the Olsson model.

The main difference to the Olsson model is the choice of the branching number, with b=3 in the first disaggregation step. This selection results in a cascade of disaggregation steps as follows: 24h->8h->4h->2h->1h->30min->15min->7.5min. This structure offers the advantage of commencing with a daily time step and achieving precisely 1-hour time steps, and later the 5 min by a linear transformation of the 7.5 min time steps. The use of b=3 in the initial disaggregation steps involves three weights for distributing the rainfall amount, as opposed to the two weights used in the Olsson (1998)-model, which maintains b=2 throughout. In the Olsson model, to achieve 1-hour time steps, the disaggregation progresses from 5.7 days -> 17 hours -> 8.5 hours -> 4.24 hours -> 2.125 hours -> 1.0625 hours (~1 hour). Conversely, employing b=2 in a cascade from a daily time step would result in: 24h -> 12h -> 6h -> 3h -> 1.5h -> 0.75h. In this scenario, a 1-hour time step is not achieved directly; instead additional modifications are required (Güntner et al., 2001, Müller und Haberlandt, 2015). As shown before by Müller and Haberlandt, with the here applied method with b=3 ("uniform splitting") better results can be achieved than with the "diversion" approach by Güntner et al. (2001) for the temporal resolution of 1h. The application of purely b=2 throughout the disaggregation process to achieve 5min time steps is only possible with a so-called fine-graining down to a few seconds and subsequent aggregation to 5min. However, the cascade model parameters for these temporal resolutions (<1min) cannot be estimated with the data used in our study, hence this is no possible option here.

l. 184: Why 'unbounded'?

This sentence highlights the difference between a bounded and an unbounded cascade model. In an unbounded cascade model, it is assumed that the model parameters are scale-independent. This implies that a single-parameter set is used for all disaggregation levels. In contrast, the bounded model operates on the assumption that the model parameters are scale-dependent. As a result, this approach employs a distinct single-parameter set for each disaggregation step. Line 183 was modified for clarification:

**"For each disaggregation step the model uses a single-parameter set (bounded cascade model)."**

l. 186: You may consider putting this at the beginning near l. 174

Thank you for this comment, this part was placed at an earlier point in section 3.1.

l. 193: The relative fraction definition does not make sense as it stands.

Unfortunately, we are not sure what the reviewer refers to exactly with 'does not make sense'. A x/(1-x)-splitting is a mechanism for distributing the rainfall volume from one coarse time step across two finer time steps. This distribution is determined by the weights W1=x and W2=1-x. Importantly, the sum of W1 and W2 equals 1, which necessitates that the value of x must lie within the range of $0 < x < 1$. Here, x signifies the fraction of the coarse time steps allocated to the first finer time step. Hence, the relative fraction definition holds significance as stated. We have rephrased the sentence as follows:

**"The relative fraction x of the rainfall amount of the coarser time step assigned to the first time step is defined as $0 < x < 1$."**

l. 195: Is it uniform, or U-shaped? – How does it look?

In Fig. S4 the distribution function f(x) is shown. It has a U-shaped pattern.

[Figure]

Figure: Distribution function f(x) at location A within the lower volume class and the enclosed position class for the disaggregation step from 1 hour to 30 minutes.

l. 211: This sentence is awkward, and seems to bring back the confusion about the target quantity.

The sentence was confusing and will be revised:

**"The parameters of the cascade model and f(x) are estimated for each location with 5 min time series for the period 01.01.2001 – 31.12.2021. The 5 min time series are extracted from a 5 km rainfall raster aggregated from the 1 km YW data. The aggregation from the YW data to a 5 km raster was used to ensure spatial consistence of the cascade model parameters to disaggregate the climate model data which also have a 5 km spatial resolution."**

l. 218: How does your implementation compare to e.g. Bürger et al. (2019) and Pons et al. (2022) who also aim at parsimonious implementation of temperature dependency? How do the 12 parameters of (an updated version of) the original cascade model of Olsson (Willems and Olsson, 2012) compare to the 40 parameters from your S1-P1 implementation?

Pons et al. (2022) introduced temperature dependency exclusively for the zero-weight generator, only for distributing the total rainfall amount from the coarse time step in only one of the two finer time steps (1/0- and 0/1-splitting). In contrast, our study incorporates temperature dependency to all cascade model parameters. Another distinction lies in the structure of the cascade model. Pons et al. used a branching number of b=2 for all disaggregation steps, resulting in a final time step of 5.625 minutes, which is no suitable for e.g. subsequent rainfall-runoff model applications. In our cascade model, we employ a branching number of b=3 in the initial step, with a 7.5-minute time step in the final disaggregation step. This 7.5-minute time step is further evenly divided into three 2.5-minute time steps, followed by an aggregation of two non-overlapping time steps (suitable for subsequent rainfall-runoff model applications).

The disaggregation model used by Bürger et al. (2019) is a simpler model compared to the model by Müller and Haberlandt (2018). A difference is the scale dependency of the parameters. Although e.g., the slope parameter $c_2$ in the model by Bürger et al. (2019) is a scale-dependent parameter, it is 'counted' as only one parameter and not

counted per scale. Comparable parameters in the model presented in our study are counted per applied scale n, which leads to a higher total parameter number. So, for a fair comparison of parameter numbers the scale-dependent parameters in Bürger et al. (2018) should be counted scale-dependent, e.g. the parameter should be counted n*c2 times instead of only once.

l. 228: event-connecting systematic is an awkward term.

We have rephrased the term to „event-cohesive".

l. 228-230: This should be removed. Unless being unity, the probabilities do not imply anything about the connectedness of the disaggregated events.

We have included an additional time step numbering in Fig. 3 to better demonstrate the connectedness of the disaggregated events. We have modified for clarification:

**"A connected disaggregation event can only occur if an enclosed time step ($Z_{1,3}$) is disaggregated with P(x/(1-x) into two wet time steps. For an enclosed time step the probability of P(x/(1-x) is very high. If a time step is disaggregated in the Starting Position Class ($Z_{1,2}$) and the next time step is enclosed ($Z_{1,3}$), there is a high probability of a 0/1-spilting, whereby the disaggregation event remains connected."**

[Figure]

Figure: Disaggregation of a wet time step ($Z_{1,2}$ & $Z_{1,4}$) to describe the similarities of probability parameters in the start and end position class (for the same volume class) for continuous rainfall events (A) and non-continuous rainfall events (B).

Table 4: This should go to the results section (# of T classes)

We would prefer not to relocate this table to the results section. This table functions as an overview of the quantity of cascade model parameters within each model variant. Therefore, the table primarily serves as a summary of the different model approaches and is not considered as a result.

Section 3.4: Consider removing this as NHESS readers are most likely familiar with the basic statistics.

We prefer to not remove section 3.4. The relative Error rE and mean Error mE are calculated over all locations and for each location over all 30 realizations. This may not be understandable for every reader. In addition, in this section the validation of the extreme values is described. The extreme value analysis is the focus of the study, so it is useful to have a brief description of the extreme values analysis

Table 5: For intensity, an 25% error in the mean and 50% error in stddev is quite something. Are future projections reliable given such errors?

The reviewer refers to Table 5. It is important to note that the cascade models leading to these results were not used for the disaggregation of the climate model data concerning future extreme values. For the analysis of future extreme values, a cascade model with temperature-dependent parameters was used. The results for this are shown in Table 7. For the rainfall intensity, the mean rE is 20 % and the mean standard deviation of the rE is 22 %. As this is a mean rE over all 45 locations, there will be locations with a lower rE and higher rE. At the locations with a lower rE (rE < 20%) the future projections are more reliable compared to locations where the rE is higher. This should be taken into account for the interpretation of the results from the climate model data. However, the overestimation of the intensity is a bias which is assumed to be stationary, so it will hold for C20 as for NTF and LTF. With the focus on extreme values, the error is smaller and reduced due to the introduction of the temperature-dependency. Nevertheless, since absolute values can be questioned, we decided to quantify the change of extreme values, which can then be applied to observed data.

Figure 4: Are you only counting events on wet days in all cases?

Only wet days are considered for the analysis of the $q_{99.9}$.

l. 327: Are you describing here the exponential dependency?

Yes, the temperature dependency is not as high for lower temperatures compared to high temperatures. For example, the difference of rainfall intensity between the lowest three temperature classes (Diff is 0.4 mm/5min for T<13°C) is not as high as between the highest classes (Diff is 5.2 mm/5min for T>13°C).

l. 330: If you only analyze wet days in Fig. 4, it may well be that the total number of high rainfall events does not occur at the highest temperatures.

The reviewer is right, in Fig. we only analyze wet days. However, in Tab. 6 we identified the 20 highest rainfall events and the temperature class in which they occur showing that the highest rainfall events do not solely occur in the highest temperature class.

Fig. 6: While the modeling error is certainly interesting, I would have found it more illuminating to have Fig. 4 repeated with T dependency (overlayed). Could that be done?

In the Fig. below the cascade model approach with temperature dependency (S1-P1-TD) has been added. Since we also think that this figure is very interesting, we propose to included it in the manuscript with the following text:

**"In Fig. 5 the impact of temperature-dependency on the $q_{99.9}$ is shown in detail for the location A-E. Notably, the exponential behavior over all temperature classes can be represented with the temperature-dependent disaggregation, which was not possible before. While for the highest temperature class (>18 °C), all model variants without temperature dependency show a great underestimation of $q_{99.9}$. S1-P1-TD leads to a better representation. A slight overestimation of $q_{99.9}$ for the highest temperature class is identified in general, but also strong overestimations are possible (location C). In the lower temperature classes (≤18 °C) there are also overestimations of $q_{99.9}$."**

[Figure]

Figure: Impact of temperature-dependency and parameter reduction on the $q_{99.9}$ rainfall intensity [mm/ 5min] for the disaggregated and the observed (quasi-gauge-adjusted radar data) time series in the temperature classes for selected locations (A-E) and the daily mean temperature.

l. 411: Is it justified to apply the daily climate models data uncorrected?

Using uncorrected climate model data would be unjustified. But as explained in the general replay the climate data used in this study has been bias-adjusted.

l. 487: The term 'physical extension' should be avoided and replaced with, e.g., physically inspired extension' or similar.

The term 'physical extension' has been replaced with **'physically inspired extension'**.

l. 514: Fortran code can be shared like any other code. So please consider sharing the code as well.

The Fortran code contains a few more extensions, which are not published yet and still under development, hence we prefer to share only the executable file. However, if someone is interested in a certain part of the code we would be happy to share a 'clean' version of it. We will try to share the Fortran code as far as possible. We have rephrased the line to:

**"..and can be shared on request."**

References:

Bürger, G., Pfister, A., and Bronstert, A.: Temperature-driven rise in extreme sub-hourly rainfall, Journal of Climate, 32, 7597–7609, 2019.

Hänsel, S., Brendel, C., Fleischer, C., Ganske, A., Haller, M., Helms, M., Jensen, C., Jochumsen, K., Möller, J., Krähenmann, S., Nilson, E., Rauthe, M., Rasquin, C., Rudolph, E., Schade, N., Stanley, K., Wachler, B., Deutschländer, T., Tinz, B., Walter, A., Winkel, N., Krahe, P., and Höpp, S.: Vereinbarungen des Themenfeldes 1 im BMVI-Expertennetzwerk zur Analyse von klimawandelbedingten Änderungen in Atmosphäre und Hydrosphäre, https:// https://doi.bafg.de/BfG/2020/ExpNHS2020.2020.01.pdf, 2020.

Pons, V., Benestad, R., Sivertsen, E., Muthanna, T. M., and Bertrand-Krajewski, J.-L.: Forecasting green roof detention performance by temporal downscaling of precipitation time-series projections, Hydrology and Earth System Sciences, 26, 2855–2874, https://doi.org/10.5194/hess-26-2855-2022, 2022.

Willems, P. and Olsson, J. (Eds.): Impacts of Climate Change on Rainfall Extremes and Urban Drainage Systems, IWA Publishing, 2012.

**Review by anonymous Referee #3**

The authors use cascade modelling to disaggregate RCM rainfall at 45 locations in Germany from 1-day to 5-min resolution, in order to estimate future changes in short-duration extremes. The model is first developed and calibrated using observations. In this procedure some reduction of parameters is attained and a temperature dependency is implemented. The model is then applied to three 30-year periods of RCM ensemble output at the same 45 locations and future changes are estimated.

The challenge of estimating future changes in short-duration rainfall extremes indeed deserves to be tackled, by different complementary approaches, one of which is combining dynamical and statistical downscaling techniques as done here. The paper is quite clear, the calculations seem well performed and everything is reasonably well presented. Still there are several deficiencies that need to be resolved before publication, in my opinion.

First of all, I agree with the criticism provided by Reviewers 1 and 2. Main issues here are, in my opinion (partly copied from previous reviews):

- daily temporal and 0.11° spatial resolution does not represent the "state-of-science";

- the impact of the (substantial) biases in precipitation from these RCMs is not assessed;

- the research questions are poorly formulated.

I also share the rest of the concerns raised by Reviewers 1 and 2 and thus I do not need to repeat them here. In addition, I have the following comments.

*Reviewer #3 is gratefully acknowledged for her/his efforts and the time spend on the manuscript. In the general reply above the main concerns of the reviewer are addressed. A point-by-point reply can be found below.*

**General:**
- Some parts (mainly in the introduction and methods sections) are overly wordy, as also commented by the other reviewers, and more or less repeat what is found in other papers on this topic. I suggest to keep it more compacts (but still stand-alone, of course) and refer to other papers for more information, when need.

*Some sections of the manuscript have already been revised with the help of specific comments by the reviewers. However, we will try to further minimize the wordiness in the revised manuscript.*

- It would be interesting to see the difference between data disaggregated from the YW data and RCM data in the C20 period, respectively. Even if the periods differ they supposedly represent present climate and one wonders about the realism in the RCM-based disaggregation.

In the Fig. below rainfall characteristics are presented for climate periods representing the present climate, using data from the YW data set (observation period, 2001-2021) and RCM data (C20 period, 1971-2000) at location D. The results of the RCM-based disaggregation show values for wet spell duration, wet spell amount, and dry spell duration that closely align with the observed values derived from the YW dataset. However, it's essential to be cautious comparing observed values to climate scenario data, as historical climate model data does not replicate the observed climate. Additionally, to analyze future climate conditions the focus is on the relative changes between climate periods rather than the absolute values. This approach adheres to the guidance provided by the German Weather Service for handling climate model data. In this study, we have taken these considerations into account, presenting future changes in extreme values as relative changes concerning the C20 period.

[Figure]

Figure: Difference between YW data set (observation period, 2001-2021) and RCM data (C20 period, 1971-2000) at location D for different rainfall characteristics. The range in the RCM data results from the minimum and maximum value of the climate ensemble.

- The Conclusions is in my opinion a Summary. Conclusions need more of…well, conclusions; what do the results mean in a broader context, how can they be used, what research remains, etc.

The reviewer is right about the current conclusion section. We propose renaming the section to "Summary and conclusion". Furthermore, we would like to offer more specific conclusions that provide valuable insights:

- Rainfall disaggregation proves to be a valuable tool for enhancing the temporal resolution of climate scenario data. This approach allows for a analysis of finer temporal rainfall patterns and their impacts.
- The methodology outlined in this study, particularly the disaggregation process, is not limited to the specific climate model data used. It can be applied to other climate scenario data, including the disaggregation of e.g. 3 h climate model data. This expansion of applicability has the potential to further enhance the accuracy of climate model results and research studies.
- The high temporal resolution future rainfall time series generated through this method offers significant potential for various applications, e.g. rainfall-runoff modeling, urban hydrological models und hydraulic models. The accuracy and effectiveness of these models depend on the temporal resolution, especially at small temporal and spatial scales.

**Specific:**

- L51: Berne et al. needs a year.

The year has been added.

- 145: To have just one dry time step (i.e. 5 dry minutes) separating wet spells is very unusually short, which also make the wet spells very short (around 20 min) and small (around 0.5 mm). I recommend a separation of hours to really encapsulate the full events. Maybe this is one reason for the quite weak performance in terms of spell durations (Table 6).

According to Dunkerley (2008), rainfall events are defined as having a minimum of one dry time step both before and after the occurrence of rainfall. In the context of 5-minute rainfall time series analysis, a dry time step is specifically defined as a period with a rainfall intensity of 0 mm/5 min. This consistent definition of rainfall events allows for direct comparisons with the findings of e.g. Müller & Haberlandt (2018) or Derx et al. (2023). In our study, we've adhered to the same definition for wet spell amount and wet spell duration analyzing 5-minute rainfall time series.

The Tab. below shows the rE [%] of continuous rainfall characteristics with rainfall events separated by 1 h dry time steps as proposed by the reviewer. Compared with the results in Tab. 5 in the manuscript a worthening of the wet spell amount and dry spell duration is notably. The difference between the model variants is comparable to the results from our study. As there are no improvements of the disaggregation results, we suggest to continue to use the one dry time step definition of rainfall events. In addition to the quite weak performance in terms of wet spell durations, the wet spell duration also strongly depends on the threshold applied for considering a time step as wet or dry (e.g. Müller and Haberlandt, 2018, Fig. 5 vs. Fig. 6). As shown by Pidoto et al. (2022) in Fig. S1, the variation of the threshold influences the resulting statistics significantly.

For the rainfall extreme events, we have chosen minimum of 4 hours without to ensure independency. This criterion aligns with the requirements outlined in the German guideline DWA 351 for the analysis of rainfall extremes.

Table: Relative Error (rE) [%] of continuous rainfall characteristics between disaggregated and observed time series for rainfall time steps (mean across 45 stations) with rainfall events separated by 1 h dry time step.

| Rainfall characteristic | rE [%] | | | | | |
|---|---|---|---|---|---|---|
| | S0-P0 | S1-P0 | S0-P1 | S1-P1 | S2-P0 | S2-P1 |
| *Wet spell duration [min]* | | | | | | |
| Mean | -30 | -28 | -30 | -28 | -4 | -3 |
| Standard deviation | -42 | -41 | -42 | -41 | -20 | -20 |
| *Rainfall intensity [mm/5 min]* | | | | | | |
| Mean | 24 | 23 | 24 | 23 | 19 | 19 |
| Standard deviation | 52 | 51 | 52 | 51 | 45 | 50 |
| *Wet spell amount [mm]* | | | | | | |
| Mean | -30 | -30 | -30 | -30 | -26 | -26 |
| Standard deviation | -32 | -32 | -33 | -32 | -24 | -23 |
| *Dry spell duration [min]* | | | | | | |

| | | | | | | |
|---|---|---|---|---|---|---|
| Mean | -30 | -30 | -30 | -30 | -28 | -28 |
| Standard deviation | -30 | -32 | -30 | -32 | -34 | -34 |

- L167: Here NTF is written as 2051-2070, later in the paper it is 2021-2050 (e.g Fig. 8). Which is correct?

The correct time period for NTF is 2021-2050. The wrong period has been corrected.

- L194: Which f(x) is used here?

f(x) is an empirical distribution function. The sentence will be revised:

**"Considering x as a random variable for all disaggregation steps, an empirical distribution function f(x) is estimated from the observed time series."**

- L379: od -> of

Thanks, it has been corrected.

- L385: Delete the first "temperature".

The first "temperature" has been deleted.

References:

Bürger, G., Pfister, A., and Bronstert, A.: Temperature-Driven Rise in Extreme Sub-Hourly Rainfall, Journal of Climate, 32, 2019.

Derx, J., Müller-Thomy, H., Kılıç, H. S., Cervero-Arago, S., Linke, R., Lindner, G., Walochnik, J., Sommer, R., Komma, J., Farnleitner, A. H., and Blaschke, A. P.: A probabilistic-deterministic approach for assessing climate change effects on infection risks downstream of sewage emissions from CSOs, Water Research, 120746, https://doi.org/10.1016/j.watres.2023.120746, 2023.

Dunkerley, D.: Identifying individual rain events from pluviograph records: a review with analysis of data from an Australian dryland site, Hydrol. Process., 22, 5024–5036, https://doi.org/10.1002/hyp.7122, 2008.

Hänsel, S., Brendel, C., Fleischer, C., Ganske, A., Haller, M., Helms, M., Jensen, C., Jochumsen, K., Möller, J., Krähenmann, S., Nilson, E., Rauthe, M., Rasquin, C., Rudolph, E., Schade, N., Stanley, K., Wachler, B., Deutschländer, T., Tinz, B., Walter, A., Winkel, N., Krahe, P., and Höpp, S.: Vereinbarungen des Themenfeldes 1 im BMVI-Expertennetzwerk zur Analyse von klimawandelbedingten Änderungen in Atmosphäre und Hydrosphäre, https:// https://doi.bafg.de/BfG/2020/ExpNHS2020.2020.01.pdf, 2020.

Müller, H. and Haberlandt, U.: Temporal rainfall disaggregation using a multiplicative cascade model for spatial application in urban hydrology, Journal of Hydrology, 556, 847–864, https://doi.org/10.1016/j.jhydrol.2016.01.031, 2018.

---

## Referee Report (RR1)

The authors have partly addressed my comments. Some of the changes or clarifications have led to additional enquiries or issues, which are listed in the following:

**Major comments:**

1) I am still not convinced that the chosen climate simulations ("DWD core ensemble") are suitable for this analysis.

   a. Convective processes are parametrized in the 0.11° simulations. However, the study tries to assess changes in 5-min rainfall and rainfall extremes, whereby the extremes in Germany are mostly governed by convective processes at this time scale. The authors argue not to use available convection-permitting model simulations, as these simulations are not available as ensemble. However, they could include such analysis to compare the results and evaluate potential deviations of convection-permitting versus parametrized setups.

   b. The authors claim that the bias-adjustment is a major advantage of the data set. However, they do not discuss or reflect on the associated issues of such a bias adjustment for their study. Which kind of quantile mapping is applied? How are extreme values handled therein? Are trends preserved? Please clarify that temperature and precipitation are adjusted independently. This is from my perspective a major issue for the "physics inspired" temperature-dependent disaggregation, and needs to be discussed as well. Adjusting precipitation and temperature independently breaks the climate-model inherent physics. I'd suggest to analyse the dependence structure of daily precipitation and temperature for the observations and the climate models during the reference period (C20). This analysis should also explicitly address rainfall extremes.

   The bias adjustment largely governs the analysed output at the daily resolution, and its implications and limitations need to be well understood by the authors and carefully presented to the reader.

   The authors claim that they are mostly interested in the climate change induced changes – why is bias adjustment needed then?

   c. The DWD core ensemble provides daily resolution only. The aim of the whole study is the analysis of 5-min and 1-h rainfall. From my perspective, the disaggregation from daily to 5-min resolution induces larger methodological uncertainty than a disaggregation from hourly climate model output to 5-min resolution.

   d. For extreme rainfall, the authors have not shown the suitability of the DWD core ensemble. They assume that the bias adjustment has led to a proper representation of extremes, however the adjustment of extremes is not straightforward due to the limited sample size.

2) In L520, the authors argue: "The key assumptions for the application of cascade models for the disaggregation of future climate model data is that the scaling behaviour of rainfall remains stationary, which is not questioned to the authors knowledge."
   If this assumption is key, the authors should not assume it on the basis that they are not

aware of any other studies that call it into question. They should substantiate the assumption themselves. For example, they could investigate the stationarity of the respective scaling based on convection-permitting simulations (e.g. https://esgf.dwd.de/projects/dwd-cps/cps-scen-v2022-01 ). Or alternatively, they should refer to other peer-reviewed studies, which show that this scaling is stationary under strong climate change.

3) L312: As far as I understand the procedure, you are using the plotting position formula. Assuming 30-year periods, you include 2.4*30 events = 72 events. You analyse up to 10-year return levels. Is that the intensity of the 3rd most intense event? Or in between the 3rd and 2nd most extreme? How do you handle extreme value statistical uncertainties?
Following up on this calculation: L475: Can you show return level return period plots for the disaggregated 5min and 1h extremes? Can you provide a map to present the spatial pattern of the return levels? Does it follow the topography (as in KOSTRA) or is it more chaotic as in RADOLAN (see Fig R1). Can you provide an evaluation of the disaggregated 2-year and 10-year return levels of 5min and 1h for the reference period (C20) compared to official rainfall guidelines return levels from KOSTRA-DWD?
(https://www.dwd.de/DE/leistungen/kostra_dwd_rasterwerte/kostra_dwd_rasterwerte.html). A successful evaluation would make a strong argument that your disaggregation procedure works for rainfall extremes for the reference period.

[Figure]

*Figure R1: 1-hourly 20-year return levels based on radar (left) and station data (right). Taken from Winterrath et al., 2017:*

Winterrath T, Brendel C, Hafer M, Junghänel T, Klameth A, Walawender E, Weigl E and Becker A 2017 Erstellung einer radargestützten Niederschlagsklimatologie (= Berichte des deutschen Wetterdienstes 251). Offenbach, Selbstverlag des Deutschen Wetterdienstes.

**Minor comments:**

The authors have addressed the majority of the minor comments of the previous review sufficiently.

L145: I'd still prefer a readable elevation legend in Fig. 1 instead of the squeezed micro-colorscale.

L175ff.: The bias adjustment is not explained sufficiently. See major comment 1b.

How is the RCM-inherent drizzle handled?

L312: Plotting positions / extreme value analysis: see major comment 3)

---

## Author Response (AR2)

**Review by Benjamin Poschlod, Referee #1**

The authors have partly addressed my comments. Some of the changes or clarifications have led to additional enquiries or issues, which are listed in the following:

We thank Benjamin Poschlod for his useful and constructive comments and the time, he spent on the manuscript. A point-by-point reply can be found below. For a better overview, the reviewers' comments are shown in black, our replies in blue and proposed changes in the manuscript bold and blue.

**Major comments:**

1) I am still not convinced that the chosen climate simulations ("DWD core ensemble") are suitable for this analysis.
    a. Convective processes are parametrized in the 0.11° simulations. However, the study tries to assess changes in 5-min rainfall and rainfall extremes, whereby the extremes in Germany are mostly governed by convective processes at this time scale. The authors argue not to use available convection-permitting model simulations, as these simulations are not available as ensemble. However, they could include such analysis to compare the results and evaluate potential deviations of convection-permitting versus parametrized setups.

The reviewer remains unconvinced regarding the suitability of the DWD core ensemble (CE) for our analysis. To check the plausibility of our results, we compared them with the convection-permitting climate model (CPM) provided by the DWD, as recommended by the reviewer. Specifically, we compared rainfall extreme values with return periods of T=2 yrs and T=10 yrs for a duration of D=1 h between the CE and the CPM for the station subset A-E in the long-term future (2071-2100) (Fig. 1). A comparison is only for long-term future possible, as the near-term future covers a different period than in our study and for C20 no data was available via the link provided by the reviewer. In addition, the CPM data had to be aggregated to match the spatial resolution of the DWD core ensemble. Our analysis demonstrates that the extreme rainfall values obtained from the CPM fall within the range of those from the CE, thereby validating the plausibility of our results.

However, we did not include this comparison in our study in order to limit the results on the analysis of the DWD core ensemble only. The main focus of our study are rainfall extreme values with a subhourly temporal resolution. The CPM data has an hourly temporal resolution and therefore offers only small value in checking the plausibility of hourly rainfall extreme values. However, in our study we focus only on relative changes in rainfall extreme values between the C20 and the future. We have refrained from presenting absolute values in our study, as climate scenario analyses priorities relative changes above absolute changes and values. Relative changes for the future in CPM 20 could not be analysed as the C20 period was not available.

[Figure]

Figure 1: Comparison of rainfall extreme values with a return period of T=2 yrs and T=10 yrs for a duration of D=1 h for the disaggregated DWD core-ensemble (CE) and the convection-permitting climate model (CPM) proposed by the reviewer for the station subset A-E in the long-term future (2071-2100).

> b. The authors claim that the bias-adjustment is a major advantage of the data set. However, they do not discuss or reflect on the associated issues of such a bias adjustment for their study. Which kind of quantile mapping is applied? How are extreme values handled therein? Are trends preserved? Please clarify that temperature and precipitation are adjusted independently. This is from my perspective a major issue for the "physics inspired" temperature-dependent disaggregation, and needs to be discussed as well. Adjusting precipitation and temperature independently breaks the climate-model inherent physics. I'd suggest to analyse the dependence structure of daily precipitation and temperature for the observations and the climate models during the reference period (C20). This analysis should also explicitly address rainfall extremes.

> The bias adjustment largely governs the analysed output at the daily resolution, and its implications and limitations need to be well understood by the authors and carefully presented to the reader.

> The authors claim that they are mostly interested in the climate change induced changes - why is bias adjustment needed then?

As pointed out in our last reply, the bias-adjustment of the DWD core ensemble is outside of the scope of our study. We used bias-adjusted and spatial downscaled climate scenario data to analyse realistic climate scenario data sets in our study and as Fig. 1 shows, we can generate plausible rainfall extreme values with the bias-adjusted climate data.

Nevertheless, we provide in the following some details of the bias-correction. The method used for quantile mapping was the Quantile-delta change mapping (QDCM) developed by Ollson et al. (2009). QDCM involves the use of simulation data in the projection period to conserve projected changes in the quantiles of the climate projection through quantile-by-quantile adjustment.

In QDCM, both observed and modelled values (for historical and future periods) are initially sorted in ascending order and assigned to percentiles. The comparison of modelled and observed percentiles provides a specific bias (or adjustment value) for each percentile. Subsequently, historical or projected model data is then treated percentile by percentile with the corresponding adjustment value. Given that the highest temperature values, and consequently the highest percentiles, typically occur toward the end of the 21st century due to climate change, it's important to note that values at this juncture would frequently receive an adjustment value primarily tailored to high percentiles in the historical period. Conversely, a significant portion of values at the onset of the projection period (from 2006) would be adjusted with a value corresponding to a medium or low percentile in the historical period. To mitigate this issue, a "moving window approach" is employed, wherein climate projection data was sorted or adjusted for 35-year time slices, and the middle 15 years were retained.

The rainfall was adjusted univariately using QDCM. It should be noted that the QDCM method was only applied to values up to the 99.9th percentile, as rainfall amounts above the 99.9th percentile are not adequately represented in the reference and projection data. For values above the 99.9th percentile, the adjustment value was extrapolated linearly. Reviewer #1 points out that the univariate QDCM is a violation of the 'physics' behind the RCM results. We agree, but at the scale applied for the QDCM we have the transition from physical coherent simulations (so the regional climate model) to statistical methods anyway (the multiplicative cascade model). So, the application of the CDCM is not an 'additional' violation and is taken into account by the authors as a necessity.

The possible extension of the bias-adjustment description would be:

**"For the bias-adjustment of the rainfall an univariately approach using the quantile-delta change mapping method was chosen by Hänsel et al. (2020). "**

> c. The DWD core ensemble provides daily resolution only. The aim of the whole study is the analysis of 5-min and 1-h rainfall. From my perspective, the disaggregation from daily to 5-min resolution induces larger methodological uncertainty than a disaggregation from hourly climate model output to 5-min resolution.

The reviewer is right. A disaggregation from 1h to 5 min induces smaller methodological uncertainties compared to the disaggregation from 1d to 5min. Nevertheless, we have chosen this approach to analyse multiple GCM-RCM combinations to provide a range of possible future extreme values rather than choosing a single GCM-RCM combination.

Also, it's important to mention that the disaggregation from 1 d to subhourly time series is an established method with acceptable uncertainties and has been carried out in several studies (e.g. Bürger et al. (2019), Müller and Haberlandt (2018), Müller-Thomy (2020), Derx et al. (2023), Pidoto et al (2022)). However, in the "Summary and Conclusion" section we pointed out that the disaggregation is a tool and can be applied on other climate data with finer temporal resolutions.

> d. For extreme rainfall, the authors have not shown the suitability of the DWD core ensemble. They assume that the bias adjustment has led to a proper representation of extremes, however the adjustment of extremes is not straightforward due to the limited sample size.

It remains unclear for the authors how the 'adjustment of extremes is not straightforward due to limited sample size'. From our understanding the bias-correction was done with the best available data. The assessment, bias adjustment, spatial downscaling and selection of climate projections for the DWD core ensemble were conducted by climatologists and is beyond the scope of our study (e.g. Hänsel et al., 2020). However, it is noteworthy that the DWD core ensemble represents the state-of-technic climate scenario data for Germany and is recommended by the DWD for climate scenario analysis. This information has been included in our study to highlight the credibility and relevance of the dataset used. Again, if reviewer #1 can provide another climate model data set with different bias-corrected GCM-RCM combinations, we will add it in the outlook.

> 2) In L520, the authors argue: "The key assumptions for the application of cascade models for the disaggregation of future climate model data is that the scaling behaviour of rainfall remains stationary, which is not questioned to the authors knowledge." If this assumption is key, the authors should not assume it on the basis that they are not aware of any other studies that call it into question. They should substantiate the assumption themselves. For example, they could investigate the stationarity of the respective scaling based on convection-permitting simulations (e.g. https://esgf.dwd.de/projects/dwd-cps/cps-scen-v2022-01). Or alternatively, they should refer to other peer-reviewed studies, which show that this scaling is stationary under strong climate change.

We agree with the reviewer. The assumption we made is quite important for our study and should be verified beforehand. To do this, we used the data from the station Bochum, which is used in the supplement of our study.

The station Bochum (N51.5026 °, E7.2289 °) has a time series length of 45 years, consisting of the periods 1940-1959, 1979-1993 and 2008-2017. This time series is not part of the data set used in the accompanied manuscript, but was selected due to its time series length. Based on the annual temperature the time series was split into two time series, one with colder years and the other one with warmer years. The threshold to classify in cold and warm years was an annual mean temperature of 10.5 °C. The two resulting times series have a length of 21 years for $condition_{cold}$ and 24 years for $condition_{warm}$ each. The mean temperature difference between the two time series is 1.2 K. This temperature difference is comparable to the temperature

increase (approximately 1.2 K) from C20 (1971-200) to the NTF (2021-2050) that is projected by the climate scenario data based on RCP 4.5 across all locations in our study.

To compare the scaling behaviour of both periods, we calculated the first three moments (Fig. 2). All three moments show an almost identical scaling behaviour in both time series. For coarser temporal resolutions, the scaling difference increases. However, the difference is relatively small (<5%). This analysis serves as evidence supporting our assumption.

[Figure]

Figure 2: Probability-weighted moments of rainfall time series for the years with condition$_{cold}$ and condition$_{warm}$ for station Bochum (based on Müller and Haberlandt, 2018).

In accordance with the reviewer's recommendation, we further investigated the scaling behaviour using the convection-permitting climate model (CPM). The CPM provides data with an hourly temporal resolution, and through the link provided by the reviewer, only the near-term future (NTF) and long-term future (LTF) data were accessible. Consequently, we were able to assess the scaling behaviour between NTF and LTF only at hourly and daily resolutions (Fig. 3). Notably, the scaling behaviour observed in both time periods is highly

similar. While there are slight variations in scaling behaviour for coarser temporal resolutions, the differences remain negligible. Thus, our assumption remains valid, supported by this additional analysis.

[Figure]

Figure 3: Probability-weighted moments of rainfall time series for the near-term future (NTF) 2031-2060 and long-term future (LTF) 2071-2100 for station Mühldorf with climate scenario data from convection-permitting climate model.

We propose to add the following text in the manuscript and the scaling behaviour analysis for station Bochum to the supplementary material section of the paper:

**"The key assumptions for the application of cascade models for the disaggregation of future climate model data is the stationary scaling behaviour of rainfall, which was proven in the supplementary material with additional data."**

3) L312: As far as I understand the procedure, you are using the plotting position formula. Assuming 30-year periods, you include 2.4*30 events = 72 events. You analyse up to 10-year return levels. Is that the intensity of the 3 rd most intense event? Or in between the 3rd and 2nd most extreme? How do you handle extreme value statistical uncertainties? Following up on this calculation: L475: Can you show return level return period plots for the disaggregated 5min and 1h extremes? Can you provide a map to present the spatial pattern of the return levels? Does it follow the topography (as in KOSTRA) or is it more chaotic as in RADOLAN (see Fig R1). Can you provide an evaluation of the disaggregated 2-year and 10- year return levels of 5min and 1h for the reference period (C20) compared to official rainfall guidelines return levels from KOSTRA-DWD?
(https://www.dwd.de/DE/leistungen/kostra_dwd_rasterwerte/kostra_dwd_rasterwerte.html)
A successful evaluation would make a strong argument that your disaggregation procedure works for rainfall extremes for the reference period.

[Figure]

*Figure R1: 1-hourly 20-year return levels based on radar (left) and station data (right). Taken from Winterrath et al., 2017:*

Winterrath T, Brendel C, Hafer M, Junghänel T, Klameth A, Walawender E, Weigl E and Becker A 2017 Erstellung einer radargestützten Niederschlagsklimatologie (= Berichte des deutschen Wetterdienstes 251). Offenbach, Selbstverlag des Deutschen Wetterdienstes.

The rainfall extreme values were analysed according DWA-A 531. The empirical return periods T were estimated with the formula given in our study:

$$T = \frac{L+0.2}{k-0.4} \cdot \frac{M}{L}$$

, where L is the number of rainfall events that is considered to be 2.4 time the length of the analysed time series number in years (M) and k the running index of the sample sorted by size. The rainfall intensities assigned to return periods $d_R$(T) were identified with the exponential distribution function:

$$d_R(T) = u + w \cdot \ln(T)$$

, where u and w are parameters determined by linear regression plotting dR(T) against ln(T). u corresponds to the y-axis value (dR) at ln(T)=0 and w is the slope of the linear regression.

A rainfall extreme intensity with a return period of T=10yrs is between the third and fourth most extreme rainfall intensities.

Regarding the second part of the reviewers comment, we are uncertain about the reviewers intentions. We have validated our extreme values from the disaggregated time series with observed values as presented in Fig. 8 of our study. We believe that a comparison of rainfall extreme values from climate scenario data with KOSTRA data is not plausible, since KOSTRA is a point dataset while the climate scenario data is a raster dataset. This difference in spatial resolution makes a comparison of extreme values impractical. Additionally, creating a map of rainfall extreme values for Germany is not feasible, as our analysis was focused on 45 specific locations within Germany. However, we have illustrated the changes in rainfall extreme values at these locations on a map of Germany in Fig. 11 of our study.

To clarify our approach regarding the calculation of return periods and rainfall extreme values we propose the following change to L312:

"First, the return periods of rainfall extremes values of the disaggregated time series are analysed. Therefore, empirical return periods (T) are estimated according to the German guideline DWA-531(DWA-531, 2012):

$$T = \frac{L+0.2}{k-0.4} \cdot \frac{M}{L} \qquad (4)$$

, where L is the number of rainfall events that is considered to be 2.4 times the length of the analysed time series number in years (M) and k the running index of the sample sorted by size. The rainfall intensities assigned to return periods $d_R(T)$ were identified with the exponential distribution function:

$$d_R(T) = u + w \cdot \ln(T) \qquad (5)$$

, where u and w are parameters determined by linear regression plotting $d_R(T)$ against ln(T) using (4). The return period allows a validation of the most extreme rainfall events."

**Minor comments:**

The authors have addressed the majority of the minor comments of the previous review sufficiently.

L145: I'd still prefer a readable elevation legend in Fig. 1 instead of the squeezed micro-colorscale.

We would propose to change the elevation legend in Fig.1 to:

[Figure]

Figure 4: Revised elevation legend in Fig. 1 of the manuscript.

L175ff.: The bias adjustment is not explained sufficiently. See major comment 1b.

How is the RCM-inherent drizzle handled?

As explained above the bias-adjustment of the DWD core ensemble is not in the scope of our study. In our study we have shortly described and mentioned the methods of bias-adjustment. As proposed by the reviewer, we have added the exact quantile mapping method that was in the bias-adjustment of the rainfall data. In addition, we have referred to literature where the bias-adjustment of the DWD core ensemble is explained.

L312: Plotting positions / extreme value analysis: see major comment 3)

Please see our comment and proposed change in the major comment 3).

References:

Bürger, G., Pfister, A., and Bronstert, A.: Temperature-driven rise in extreme sub-hourly rainfall, Journal of Climate, 32, 7597–7609, 2019.

Derx, J., Müller-Thomy, H., Kılıç, H. S., Cervero-Arago, S., Linke, R., Lindner, G., Walochnik, J., Sommer, R., Komma, J., Farnleitner, A. H., and Blaschke, A. P.: A probabilistic-deterministic approach for assessing climate change effects on infection risks downstream of sewage emissions from CSOs, Water Research, 120746, https://doi.org/10.1016/j.watres.2023.120746, 2023.

Hänsel, S., Brendel, C., Fleischer, C., Ganske, A., Haller, M., Helms, M., Jensen, C., Jochumsen, K., Möller, J., Krähenmann, S., Nilson, E., Rauthe, M., Rasquin, C., Rudolph, E., Schade, N., Stanley, K., Wachler, B., Deutschländer, T., Tinz, B., Walter, A., Winkel, N., Krahe, P., and Höpp, S.: Vereinbarungen des Themenfeldes 1 im BMVI-Expertennetzwerk zur Analyse von klimawandelbedingten Änderungen in Atmosphäre und Hydrosphäre, https:// https://doi.bafg.de/BfG/2020/ExpNHS2020.2020.01.pdf, 2020.

Müller, H. and Haberlandt, U.: Temporal rainfall disaggregation using a multiplicative cascade model for spatial application in urban hydrology, Journal of Hydrology, 556, 847–864, https://doi.org/10.1016/j.jhydrol.2016.01.031, 2018.

Müller-Thomy, H.: Temporal rainfall disaggregation using a micro-canonical cascade model: possibilities to improve the autocorrelation, Hydrol. Earth Syst. Sci., 24, 169–188, https://doi.org/10.5194/hess-24-169-2020, 2020.

Olsson, J., Berggren, K., Olofsson, M., and Viklander, M.: Applying climate model precipitation scenarios for urban hydrological assessment: A case study in Kalmar City, Sweden, Atmospheric Research, 92, 364–375, https://doi.org/10.1016/j.atmosres.2009.01.015, 2009.

Pidoto, R., Bezak, N., Müller-Thomy, H., Shehu, B., Callau-Beyer, A. C., Zabret, K., and Haberlandt, U.: Comparison of rainfall generators with regionalisation for the estimation of rainfall erosivity at ungauged sites, Earth Surf. Dynam., 10, 851–863, https://doi.org/10.5194/esurf-10-851-2022, 2022.

---

## Author Response (AR3)

**Short response to the editor**

We thank Nadav Peleg for his comments and the editorial of our manuscript.

We suggest to leave Table 1 in section 2 in the main manuscript. Throughout the analyses in the manuscript, we only show relative errors. These relative errors cannot be interpreted without knowledge of the absolute values. Therefore, we would not move Table 1 to the supplement.

**Review by Benjamin Poschlod, Referee #1**

Dear authors, thank you for the revision, very interesting additional results. The additional investigations have addressed my concerns sufficiently. I only suggest some minor technical additions:

We thank Benjamin Poschlod for his useful and constructive comments and the time he spent on the manuscript. A point-by-point reply can be found below. For a better overview, the reviewers' comments are shown in black, our replies in blue and proposed changes in the manuscript bold and blue.

**Minor comments:**

- I'd add a reference in the main article to refer to the comparison to the CPM in the supplement

We have added:

**"Furthermore, the results are confirmed by a comparison with a convection-permitting climate model (see supplementary material S2 for details)."**

- Please add a statement about extreme values in the bias adjustment in the article similar to your author reply: "It should be noted that the QDCM method was only applied to values up to the 99.9th percentile, as rainfall amounts above the 99.9th percentile are not adequately represented in the reference and projection data. For values above the 99.9th percentile, the adjustment value was extrapolated linearly."

Many of the daily bias-adjusted rainfall sums above the 99.9 percentile will govern your results.

We have added:

**"However, the QDCM method is only applied to rainfall values up to the 99.9th percentile, as rainfall amounts above the 99.9th percentile are not adequately represented in the reference and projection data. For rainfall values above the 99.9th percentile, the adjustment value was extrapolated linearly."**

- I would rephrase L526: You show that the stationary scaling behavior of rainfall only for one station (Bochum), which is well enough to address my reviewer comment; however I'd not say that you "prove" this assumption generally, but "empirically show the assumption to be reasonable in the study area".

We have rephrased L526 to:

**"The key assumptions for the application of cascade models for the disaggregation of future climate model data is the stationary scaling behaviour of rainfall, which was empirically shown to be reasonable in the study area with additional data in the supplementary material."**

---

## Author Response (AR4)

Since the last Author's Response, there are no open review points from the referees. Therefore, there is no new point-by-point response and reference is made to the previous Authors' responses.